complexity/mathematical modelling

complex networks, multiplex networks, link prediction

**Author for correspondence:**
Amir Mahdi Abdolhosseini-Qomi
e-mail: abdolhosseini@ut.ac.ir

# Link prediction in real-world multiplex networks via layer reconstruction method

Amir Mahdi Abdolhosseini-Qomi, Seyed Hossein Jafari, Amirheckmat Taghizadeh, Naser Yazdani, Masoud Asadpour and Maseud Rahgozar

University of Tehran, Department of Electrical and Computer Engineering, Tehran 1439957131, Iran

 AMA-Q, 0000-0003-0085-202X; SHJ, 0000-0001-5088-7795;
AT, 0000-0001-8874-2188

Networks are invaluable tools to study real biological, social and technological complex systems in which connected elements form a purposeful phenomenon. A higher resolution image of these systems shows that the connection types do not confine to one but to a variety of types. Multiplex networks encode this complexity with a set of nodes which are connected in different layers via different types of links. A large body of research on link prediction problem is devoted to finding missing links in single-layer (simplex) networks. In recent years, the problem of link prediction in multiplex networks has gained the attention of researchers from different scientific communities. Although most of these studies suggest that prediction performance can be enhanced by using the information contained in different layers of the network, the exact source of this enhancement remains obscure. Here, it is shown that similarity w.r.t. structural features (eigenvectors) is a major source of enhancements for link prediction task in multiplex networks using the proposed layer reconstruction method and experiments on real-world multiplex networks from different disciplines. Moreover, we characterize how low values of similarity w.r.t. structural features result in cases where improving prediction performance is substantially hard.

## 1. Introduction

Real-world systems are made of elements with complex interconnections in between. Real-world networks like biochemical, human and air transportation networks are examples of biological, social and technological systems, respectively. Scientists have studied these systems extensively under the title of complex networks or network science [1,2].

The core concept of these researches is that the collective behaviour of the whole system is not just a simple superposition of individual behaviour of elements of the system [3]. These complex interactions lead to non-trivial behaviour of the whole system. More specifically, neurons, human beings and airports as the elements of aforementioned systems are linked by inter-cellular connections, acquaintances and flights, respectively, to shape the specific purposes of the systems [4–6].

Recently, a higher resolution image of these systems shows that the connection type between elements of a system does not confine to one type, but to a variety of connection types [7–9]. Biological studies show that inter-cellular connections can be further divided into electrical and chemical connections [10]. In a similar way, people are connected to each other as they are members of a family, friends or co-workers [5]. Also, a closer look into air transportation system reveals that flights are not operated by a single airline, but dozens of airlines form the whole system [11]. So, this new dimension of complexity may affect the behaviour of complex networks, and it deserves to be studied with scrutiny.

The first step in this study is to find an appropriate mathematical representation for these systems. Multiplex networks are a suitable way of encoding this new dimension of complexity with a set of nodes which are connected in different layers via different types of links. Each layer consists of a replica of nodes and one type of link. In the multiplex (duplex) network, a one-to-one mapping is assumed between the nodes of the two layers; in other words, both layers have the exact same set of nodes, whereas inside each layer, the wiring of the edges may differ. The process of forming this mapping is also known as layer coupling.

Real multiplex networks are not simple stacks of network layers [9]. Although there are many possibilities for coupling of layers in these networks, they are coupled in a way which is far from random coupling. This fact, known as correlated multiplexity [12], leads to empirically significant interlayer degree correlation and link overlap in real multiplex networks [9]. Functionality of a network is also affected by multiplexity as the function of one layer may affect the function of other layers, which is not additive or linear in general [9].

Network scientists have devoted a significant effort to uncover the underlying organization of real single-layer (simplex) networks [13]. Some mechanisms have already been accepted as primary driving forces in network organization, including homophily [14,15], triadic closure [16], preferential attachment [17,18] and social balance [19]. However, these mechanisms cannot provide a complete explanation of the aforementioned organization, i.e. link formation in real-world networks is usually driven by both regular and irregular factors, and only the former can be explained using mechanistic models [20]. This fact sheds light on the link prediction problem in which the set of observed links in a network is used to estimate the likelihood that a non-observed link exists [21]. The regularities of networks can be explained by models, and models provide clues about new link prediction algorithms and vice versa [22]. The extent to which the network formation is explicable coincides with our capacity to predict missing links [20].

A large portion of link prediction algorithms can be classified as similarity-based algorithms which are based on definition of structural similarity measures [21] between unconnected node pairs. A mechanism like triadic closure is the basis for success of common-neighbour-based methods in which structural similarity is defined as weighted sum of number of common neighbours [23,24]. Structural similarity measures can be very simple or very complicated, and they may work for some networks while fail for the others [21]. This means that to choose an appropriate structural similarity measure for link prediction in a specific network, prior knowledge is needed about network organization.

The challenge of link prediction in real multiplex networks is twofold. In these networks, organization of network is different but related from one layer to another. Therefore, a similarity measure is needed to determine the degree of organizational relatedness of different layers in a multiplex network. On the other hand, when it comes to multiplex networks, it is hard to extend the notion of structural similarity [25]. In the target layer with missing links, the conventional structural similarity measures reflect how much disconnected node pairs are similar from the perspective of the target layer, but it is also needed to know how much these node pairs are similar from other layers' point of view.

The main contributions of this paper are to address aforementioned challenges. Regarding the first challenge, for each layer of a multiplex network, eigenvectors are considered as the structural features [26]. Here, also a new notion of structural similarity of layers is introduced based on the assumption that two layers are similar if they share similar structural features, i.e. eigenvectors. Observation of cosine similarity of eigenvectors in real duplex networks indicates that this assumption holds and is a major source of information redundancy in real multiplex networks.

Using eigenvectors as the structural features helps in defining structural similarity measure which is free of prior knowledge of network organization. Otherwise, the appropriate structural similarity measure may differ from one layer to another and this makes reaching a unified framework more difficult. Recently, structural similarity measure based on eigenvectors of networks has been introduced [20]. This line of work, known as structural perturbation method (SPM), assumes that the missing links in a network are predictable if the removal or addition of randomly selected links of the network does not significantly change the structural features of network. In other words, in a highly predictable network, addition of missing links makes almost no change in eigenvectors as the structural features of the network and just modifies the eigenvalues. Although network organization differs across different layers of real multiplex networks, eigenvectors are good bases for definition of structural similarity in each layer. Also, extensive experiments show that SPM outperforms state-of-the-art link prediction methods in both accuracy and robustness [20], and this makes it a suitable baseline for link prediction in different layers of multiplex networks which are of different nature.

The second challenge is about extension of similarity notion to multiplex networks. Considering a target layer with missing links, a new notion of similarity is needed to reflect the similarity of unconnected node pairs in this layer from the perspective of other layers. In real-world multiplex networks, the similarity of layers w.r.t. structural features bring to mind how well one layer can be reconstructed by structural features of other layers. Formulation of this idea as an optimization problem leads to a convex optimization problem, and the globally optimum answer reveals the best possible reconstruction. The similarity of unconnected node pairs in one layer from the perspective of another layer can be reflected through the best reconstruction of the former with structural features of the latter. This method, which is referred to as layer reconstruction method (LRM), leverages this concept for link prediction task. LRM considers the unconnected node pairs in the target layer as similar if they are not only similar from the perspective of the target layer but also from the perspective of other layers. Experiments on real multiplex networks from different disciplines show that LRM benefits from information redundancy in different layers of real-world multiplex networks. The information redundancy helps the results to stay robust even under a high fraction of missing links.

The rest of the paper is organized as follows: §2 reviews the related works in the literature. In Results, §3, we elaborate on the observations, the experiments and the findings of this study. Then, we discuss the outcomes of the research in §4. Section 5 explains data, link prediction problem, evaluation metrics and structural perturbation method.

## 2. Related works

Recently, link prediction in multiplex networks has attracted the attention of researchers [27]. The geometric embedding has been used to reveal the hidden correlations in real multiplex networks [28]. These correlations have been further used for trans-layer link prediction. Trans-layer link prediction is about finding missing links in one layer using a similarity measure on another layer, and its effectiveness has been evaluated in contrast to binary link predictor which is based on edge overlap. Also, it is shown that geometric correlations are not enough to explain the high edge overlap in real multiplex networks and a link persistence factor can both improve the reproduction of edge overlap and improve performance of trans-layer link prediction [29].

Some researchers have approached the problem using feature engineering and applied machine learning. A study of a multiplex online social network demonstrates the importance of multiplex links (link overlap) in significantly higher interaction of users based on available side information [27]. The authors consider Jaccard similarity of extended neighbourhood of nodes in multiplex network as a feature for training a classifier for link prediction task. They have shown that using multiplex feature enhances the link prediction performance. A similar work on the same dataset benefits from node-based and meta-path-based features [30]. A specialized type of these meta-paths are tailored to be originated from and ending at communities. The effectiveness of the features has been examined by a binary classification for link predication task. Recently, other interlayer similarity features based on degree, betweenness, clustering coefficient and similarity of neighbours have been used [31].

Furthermore, the issue of link prediction has been investigated in a scientific collaboration on multiplex networks [32]. The authors have proposed a supervised rank aggregation paradigm to

benefit from the node pairs ranking information which is available in other layers of the network. Another study uses rank aggregation method on a time-varying multiplex network [33]. The effect of other layers on the target layer of link prediction has been measured using global link overlap. A recent work combines feature engineering and rank aggregation [34]. Two features based on hyperbolic distance are being used and link overlap is considered for relevance of layers.

The issue of layer relevance and its effect on link prediction is studied in [35]. The authors use global link overlap and Pearson correlation coefficient of node features as measures of layer relevance, and later they use it to combine the basic similarity measures of each layer. The results support that the more layers are relevant, the better performance of link prediction is attained. Moreover, layer relevance (or in other words, layer interdependence) has been investigated based on the underlying community structure of the multiplex networks [36]. The authors have identified the difference in layer interdependence of social and genetic multiplex networks and how this issue affects the performance of link prediction in these networks.

Other research communities have also tackled the link prediction problem in multiplex networks but with their own terminology. In the machine learning community, this problem is known as multi-relational learning [37]. Most of the works in this area are based on heuristic loss. Also, the factorization of network has been used for link prediction task [38]. This direction of work formulates the problem as a supervised learning.

Our work is distinct from the existing literature as it introduces a new relevance measure for layers of multiplex networks which is in agreement with our intuitions and uses eigenvectors as structural features. Moreover, the provided insight about the cases in which it is hard to improve the performance of link prediction is complementary to existing literature. As our main contribution, we propose a novel link prediction method which incorporates structural features from auxiliary layers to predict links in an arbitrary target layer that outperforms state-of-the-art methods.

# 3. Results

## 3.1. Similarity of structural features

Here, we study the similarity of layers in multiplex networks with each layer being a simple graph. One way to represent a simple graph is via the adjacency matrix that contains all information of the corresponding layer. The structure of each layer is made of some substructures of local (e.g. triads) and global (e.g. hubs) importance. Here, we state that a good layer similarity measure should compare layers based on all substructures, from local to global ones. Eigenvectors of adjacency matrix are known to be associated with substructures of networks [26,39–41]. The proposed layer similarity measure is different from the existing literature [42] as it takes various substructures into account at the same time.

Now, we introduce our method for measuring the similarity across layers of multiplex networks.

Consider a multiplex network $G$ of $N$ nodes and $M$ undirected and unweighted layers. We can associate with each layer $\alpha$, $\alpha = 1, \ldots, M$, an adjacency matrix $A^{[\alpha]} = \{\alpha_{ij}^{[\alpha]}\}$, where $\alpha_{ij}^{[\alpha]} = 1$ if node $i$ and node $j$ are connected through a link on layer $\alpha$ and $\alpha_{ij}^{[\alpha]} = 0$ otherwise. Since all layers are real and symmetric, they can be diagonalized as

$$A^{[\alpha]} = \sum_{k=1}^{N} \lambda_k^{[\alpha]} x_k^{[\alpha]} x_k^{[\alpha]^T}, \tag{3.1}$$

where $\lambda_k^{[\alpha]}$ and $x_k^{[\alpha]}$ are the eigenvalue and corresponding orthogonal and normalized eigenvector for $A^{[\alpha]}$, respectively. The eigenvectors of each layer reflect the structural features of that layer [26]. So, the similarity matrix of structural features of two layers $\alpha$ and $\beta$ can be defined as $O^{[\alpha,\beta]} = \{o_{kl}^{[\alpha,\beta]}\}$, where $o_{kl}^{[\alpha,\beta]} = |x_k^{[\alpha]^T} x_l^{[\beta]}|$, $x_k$ is the eigenvector of $k$th largest eigenvalue of $A^{[\alpha]}$, and $x_l$ is the eigenvector of $l$th largest eigenvalue of $A^{[\beta]}$. This is the absolute pairwise cosine similarity of eigenvectors of two layers and the elements of $O^{[\alpha,\beta]}$ will be bounded in the interval [0, 1]. It is also noticeable that $O^{[\alpha,\alpha]} = I$, the identity matrix, which is the case of ideal similarity of structural features. In real-world multiplex networks, the similar eigenvectors of two layers are not necessarily ordered accordingly. For example, the fifth eigenvector of layer $\alpha$ (corresponding to fifth largest eigenvalue) may be similar to the tenth eigenvector of layer $\beta$ (corresponding to tenth largest eigenvalue). Therefore, the permutation matrix of rows, $P_r$, and the permutation matrix of columns, $P_c$, can be defined in a way that maximizes the trace of $P_r O^{[\alpha,\beta]} P_c$. A good approximate solution is to find the largest element of $O$ and then deleting

the corresponding row and column from the matrix and repeating this process for $N$ times. Therefore, a simple measure for similarity of structural features can be defined as

$$q^{[\alpha,\beta]} = \frac{\mathrm{tr}(P_r O^{[\alpha,\beta]} P_c)}{N}, \tag{3.2}$$

which will be always lower than or equal to 1. If two layers have identical structural features, $q^{[\alpha,\alpha]}$ will be 1. The more similar the structural features of the layers are, the higher value of $q^{[\alpha,\beta]}$ will be.

In addition, a mechanism is needed to determine the significance of values of $q^{[\alpha,\beta]}$. This can be achieved by comparing the values of $q^{[\alpha,\beta]}$ with similarity of structural features in the null models in which one of layers $\alpha$ or $\beta$ or both of them has been replaced by a random network. Using hypothesis testing framework, the null hypothesis $H_0$ can be considered as 'the observed value of similarity of structural features in the real network is due to randomness'. The alternative hypothesis $H_A$ will be 'the observed value of similarity of structural features in the real network is due to regularities'. The null hypothesis can be rejected, i.e. the observation is significant, if it is unlikely to see $q^{[\alpha,\beta]}$ or extremer values of similarity of structural features in different realizations of the null model.

Let $g_\alpha$ and $g_\beta$ be two Erdos–Renyi graphs [43] with average link density of layers $\alpha$ and $\beta$, respectively, such that

$$g_\alpha = \mathrm{ER}\left(N, \frac{|E_\alpha|}{N(N-1)/2}\right) \tag{3.3}$$

and

$$g_\beta = \mathrm{ER}\left(N, \frac{|E_\beta|}{N(N-1)/2}\right), \tag{3.4}$$

where $g = \mathrm{ER}(n, p)$ is an Erdos–Renyi graph with $n$ vertices and $\Pr[g(i, j) = 1] = p$. The random variables which represent the similarity of structural features for different realizations of the null models are denoted by $Q^{\mathrm{LR}}$, $Q^{\mathrm{RL}}$ and $Q^{\mathrm{RR}}$, corresponding to the events $q^{[\alpha, g_\beta]}$, $q^{[g_\alpha, \beta]}$ and $q^{[g_\alpha, g_\beta]}$, respectively. Also, $p$-values can be calculated as

$$p_{\mathrm{value}} = \Pr[Q^{\mathrm{LR}} \geq q^{\alpha,\beta}], \tag{3.5}$$

$$p_{\mathrm{value}} = \Pr[Q^{\mathrm{RR}} \geq q^{\alpha,\beta}] \tag{3.6}$$

and
$$p_{\mathrm{value}} = \Pr[Q^{\mathrm{RR}} \geq q^{\alpha,\beta}]. \tag{3.7}$$

Considering the significance level of 0.05, the significance of observed $q^{[\alpha,\beta]}$ will be determined.

## 3.2. Measurement of layer similarity in real-world networks

Figure 1 demonstrates the similarity of structural features in Physicians advice/discuss network which is derived according to following steps: (i) Calculate the eigenvectors of the adjacency matrices of advice and discuss networks. (ii) Form the similarity matrix of structural features $O^{[\mathrm{advice, discuss}]}$. (iii) Find the permutation matrices $P_r$ and $P_c$. (iv) Select the top (here 10%) most similar row–column pairs of $P_r O^{[\mathrm{advice,discuss}]} P_c$. The heatmap of selected submatrix is shown in figure 1a. Using the same steps for the randomized advice ($g_{\mathrm{advice}}$) and discuss ($g_{\mathrm{discuss}}$) networks, figure 1b is generated. By visual inspection, it is clear that the value of similarity matrix trace changes significantly before and after randomization. This indicates that advice layer and discuss layers are similar w.r.t. structural features. In other words, the organization of the layers of these networks are correlated, and not totally independent from each other. For example, if triadic closure plays an important role in formation of links in advice layer, a similar criteria holds for discuss layer, too.

Beyond the visual inspection, it should be determined whether the value of similarity of structural features in advice/discuss network ($q^{[\mathrm{advice/discuss}]} = 0.383$ as shown in table 1) is statistically significant or not. There are many ways of randomizing a real network, so the value of $q^{[\mathrm{advice/discuss}]}$ differs from one realization to another, as already denoted by random variable $Q^{\mathrm{RR}}$. Assuming that $Q^{\mathrm{RR}}$ has a normal distribution, the mean and standard deviation are calculated by 50 samples. The mean value in this case is 0.186 and $p_{\mathrm{value}} = \Pr[Q^{\mathrm{RR}} \geq 0.383] = 0$. So, it can be inferred that it is very unlikely for the null model to produce such level of similarity of structural features and the observed $q^{[\mathrm{advice/discuss}]}$ is statistically significant. All the values in table 1 are calculated accordingly.

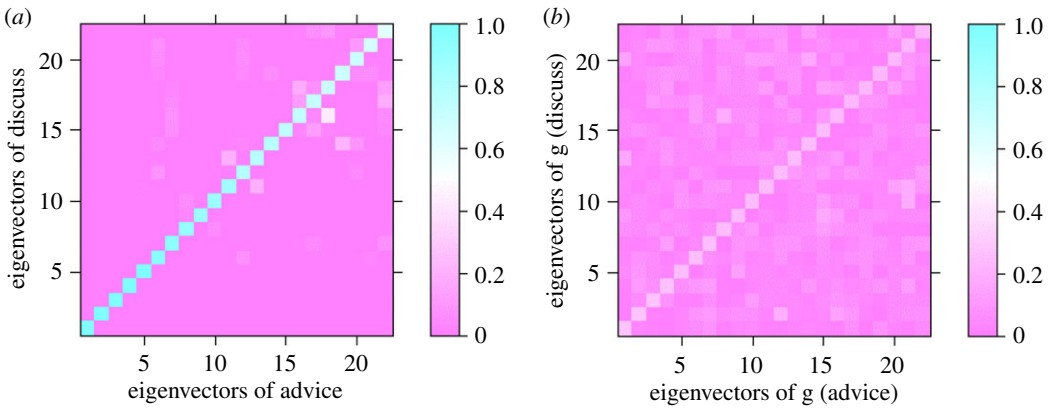

**Figure 1.** Similarity of structural features between Physicians advice/discuss layers. The heatmap of absolute cosine similarity between topmost similar eigenvectors of the two layers: (*a*) before randomization. (*b*) After randomization.

**Table 1.** The calculated values of similarity of structural features for all layer pairs (real–real) of multiplex networks under study. Also, the mean values of similarity of structural features for the null models (real–random, random–real and random–random) and the respective *p*-value which indicates the likelihood of observation of real–real value under the assumptions of the null-model is mentioned.

| multiplex name | layer pair | real–real | real–random | *p*-value | random–real | *p*-value | random–random | *p*-value |
|---|---|---|---|---|---|---|---|---|
| Physicians | advice/discuss | 0.383 | 0.191 | 0 | 0.192 | 0 | 0.186 | 0 |
| | advice/friend | 0.340 | 0.194 | 0 | 0.191 | 0 | 0.185 | 0 |
| CS-Aarhus | lunch/FB | 0.365 | 0.316 | $1.59 \times 10^{-12}$ | 0.315 | 0 | 0.318 | $8.12 \times 10^{-14}$ |
| | lunch/co-author | 0.410 | 0.369 | $7.77 \times 10^{-3}$ | 0.312 | 0 | 0.312 | 0 |
| | lunch/leisure | 0.376 | 0.320 | $3.00 \times 10^{-15}$ | 0.316 | 0 | 0.315 | 0 |
| | lunch/work | 0.378 | 0.316 | 0 | 0.317 | 0 | 0.319 | 0 |
| | FB/co-author | 0.513 | 0.456 | 0.026 | 0.328 | 0 | 0.329 | 0 |
| | FB/leisure | 0.450 | 0.334 | 0 | 0.322 | 0 | 0.317 | 0 |
| | FB/work | 0.364 | 0.314 | $1.11 \times 10^{-15}$ | 0.323 | $9.49 \times 10^{-9}$ | 0.318 | $2.58 \times 10^{-11}$ |
| | co-author/ leisure | 0.509 | 0.343 | 0 | 0.423 | $8.18 \times 10^{-4}$ | 0.337 | 0 |
| | co-author/work | 0.401 | 0.312 | 0 | 0.356 | $5.57 \times 10^{-4}$ | 0.311 | 0 |
| | leisure/work | 0.365 | 0.314 | $3.26 \times 10^{-14}$ | 0.322 | $8.37 \times 10^{-9}$ | 0.317 | $1.33 \times 10^{-13}$ |
| Brain | structure/ function | 0.312 | 0.275 | $1.44 \times 10^{-15}$ | 0.272 | 0 | 0.274 | $1.33 \times 10^{-15}$ |
| *C. elegans* | electric/ chem-mono | 0.197 | 0.175 | 0 | 0.177 | 0 | 0.175 | 0 |
| | electric/ chem-poly | 0.194 | 0.174 | 0 | 0.176 | 0 | 0.175 | 0 |
| | chem-mono/ chem-poly | 0.233 | 0.175 | 0 | 0.175 | 0 | 0.176 | 0 |
| *Drosophila* | suppress/ additive | 0.137 | 0.115 | 0 | 0.117 | 0 | 0.112 | 0 |
| Air/Train | Air/Train | 0.338 | 0.301 | $3.77 \times 10^{-15}$ | 0.303 | $1.38 \times 10^{-10}$ | 0.305 | $1.77 \times 10^{-11}$ |
| LondonTransport | tube/ overground | 0.230 | 0.238 | 0.636 | 0.201 | $5.85 \times 10^{-7}$ | 0.228 | 0.463 |
| | tube/DLR | 0.239 | 0.293 | 0.993 | 0.216 | 0.002 | 0.277 | 0.924 |
| | overground/DLR | 0.338 | 0.372 | 0.950 | 0.344 | 0.552 | 0.453 | 0.893 |

**Table 2.** Basic statistics of multiplex networks under study. A node is active in each layer if it has at least one link in that layer. Node multiplexity denotes the ratio of nodes which are active in more than one layer.

| multiplex name | no. of layers | no. of nodes | node multiplexity | layer name | no. of active nodes | no. of links |
|---|---|---|---|---|---|---|
| Physicians | 3 | 246 | 0.93 | advice | 215 | 449 |
| | | | | discuss | 231 | 498 |
| | | | | friend | 228 | 423 |
| CS-Aarhus | 5 | 61 | 0.96 | lunch | 60 | 193 |
| | | | | FB | 32 | 124 |
| | | | | co-author | 25 | 21 |
| | | | | leisure | 47 | 88 |
| | | | | work | 60 | 194 |
| Brain | 2 | 90 | 0.85 | structure | 85 | 230 |
| | | | | function | 80 | 219 |
| C. elegans | 3 | 280 | 0.98 | electric | 253 | 515 |
| | | | | chem-mono | 260 | 888 |
| | | | | chem-poly | 278 | 1703 |
| Drosophila | 2 | 839 | 0.89 | suppress | 838 | 1858 |
| | | | | additive | 755 | 1424 |
| Air/Train | 2 | 69 | 1 | air | 69 | 180 |
| | | | | train | 69 | 322 |
| LondonTransport | 3 | 368 | 0.13 | tube | 271 | 312 |
| | | | | overground | 83 | 83 |
| | | | | DLR | 45 | 46 |

The results in table 1 indicate that all layer pairs of multiplex networks under study are similar w.r.t. structural features with the exception of LondonTransport network. This means that these networks show properties that are unlikely to be seen from their randomized counterparts. The node multiplexity [9] measure sheds light on the exceptional case of LondonTransport network. Node multiplexity indicates the percentage of nodes in a multiplex network which are active (have an edge) in more than one layer. Table 2 shows the value of node multiplexity for multiplex networks under study. Clearly, the value of node multiplexity for LondonTransport is much lower than other networks. The zero node multiplexity means no node is shared among networks on different layers of a multiplex network and the structural features of different layers will be in disjoint subspaces. This condition leads to even less similarity w.r.t. structural features when compared with randomized networks in which the eigenvectors are distributed isotropically at random and span the whole space. So, the special case of LondonTransport network is justifiable and a remedy for this situation is to exclude many nodes which are only active in Tube layer.

Also for multiplex networks with more than two layers a relative comparison is possible. According to table 1, in C. elegans network two layers with chemical nature (chem-mono and chem-poly) are more similar to each other w.r.t. structural features rather than the other layer which is of electrical nature. In Physicians network, it can be inferred that the organization of discuss and friend layers are more similar to each other compared with advice layer. It has been discussed that Facebook (FB) layer is the less coverable layer in CS-Aarhus network, i.e. combining all links in the other layers only covers 0.64 of links in Facebook layer and thus this layer brings new information that is not provided in other layers [44]. Here, it can be added that the organization of Facebook layer is more similar to co-author layer and less similar to work and lunch layers. Maybe, the root cause of this observation is that work and lunch activities are bound to specific geographical locations while this is not the case for co-authorship relations in which people may cooperate with each other remotely. This gives a better understanding of the nature of Facebook relationships.

## 3.3. Layer reconstruction method

Consider two layers $\alpha$ and $\beta$ of a multiplex network with similar structural features and their adjacency matrices $A^{[\alpha]}$ and $A^{[\beta]}$, respectively. The similarity of structural features of these two layers indicates that they share some similar eigenvectors. So, the eigenvectors of $A^{[\beta]}$ can contribute in reconstruction of $A^{[\alpha]}$ as

$$\tilde{A}^{[\alpha,\beta]} = \sum_{k=1}^{N} \mu_k x_k^{[\beta]} x_k^{[\beta]^T}, \tag{3.8}$$

where $\tilde{A}^{[\alpha,\beta]}$ is the reconstruction of layer $\alpha$ by structural features of layer $\beta$, and $\mu_k$ determines the extent of contribution of each structural feature. This reconstruction should be as close as possible to $A^{[\alpha]}$, so the aforementioned contribution comes from solving the following optimization problem:

$$\min_{\mu_k} \|A^{[\alpha]} - \tilde{A}^{[\alpha,\beta]}\|_F^2, \tag{3.9}$$

where $\|.\|_F^2$ denotes Frobenius norm. Omitting the layer superscript for the sake of notational simplicity, the objective function $Z:\mathbb{R}^N \to \mathbb{R}$ can be expanded as

$$Z(\mu_1, \ldots, \mu_N) = \|A - \tilde{A}\|_F^2 = \|A\|_F^2 + \| - \tilde{A}\|_F^2 + 2 < A, \ -\tilde{A} >_F, \tag{3.10}$$

where $\langle .,. \rangle_F$ is Frobenius inner product. Since $\|A\|_F^2$ is constant w.r.t. $\mu_k$, the objective function in equation (3.10) is reducible to

$$Z(\mu_1, \ldots, \mu_N) = \|\tilde{A}\|_F^2 - 2 < A, \tilde{A} >_F. \tag{3.11}$$

The adjacency matrices are real-valued and symmetric; thus, equation (3.11) can be written as

$$Z(\mu_1, \ldots, \mu_N) = \mathrm{tr}(\tilde{A}^T \tilde{A}) - 2\mathrm{tr}(A^T \tilde{A}), \tag{3.12}$$

where $\mathrm{tr}(.)$ is the trace of matrix. Substituting $\tilde{A}$ with equation (3.8), equation (3.12) can be expanded as

$$
\begin{aligned}
Z(\mu_1, \ldots, \mu_N) &= \mathrm{tr}\Big\{(\mu_1 x_1 x_1^T + \cdots + \mu_N x_N x_N^T)^T (\mu_1 x_1 x_1^T + \cdots + \mu_N x_N x_N^T)\Big\} \\
&\quad - 2\mathrm{tr}(A^T(\mu_1 x_1 x_1^T + \cdots + \mu_N x_N x_N^T)) \\
&= \mathrm{tr}\left(\sum_{k=1}^{N} \mu_k^2 x_k x_k^T x_k x_k^T + \sum_{k=1}^{N} \sum_{l=1, l \neq k}^{N} \mu_k \mu_k x_k x_k^T x_l x_l^T\right) \\
&\quad - 2\mathrm{tr}(A^T(\mu_1 x_1 x_1^T + \cdots + \mu_N x_N x_N^T)).
\end{aligned} \tag{3.13}
$$

Knowing eigenvectors are normal and orthogonal, $x_k^T x_K = 1$ and $x_k^T x_l = 0$, $k \neq l$. Subsequently, equation (3.13) can be simplified as

$$
\begin{aligned}
Z(\mu_1, \ldots, \mu_N) &= \mathrm{tr}\left(\sum_{k=1}^{N} [\mu_k^2 x_k x_k^T - 2\mu_k A^T x_k x_k^T]\right) \\
&= \sum_{k=1}^{N} \mathrm{tr}(\mu_k^2 x_k x_k^T - 2\mu_k A^T x_k x_k^T) \\
&= Z(\mu),
\end{aligned} \tag{3.14}
$$

where $\mu = (\mu_1, \ldots, \mu_N)$. To solve the optimization problem, the first derivatives of objective function should be set to zero, $\partial Z(\mu)/\partial \mu = 0$. So for $\mu_k$, the equation will be

$$
\begin{aligned}
\frac{\partial Z(\mu)}{\partial \mu_k} &= \sum_{k=1}^{N} \frac{\partial}{\partial \mu_k} \mathrm{tr}(\mu_k^2 x_k x_k^T - 2\mu_k A^T x_k x_k^T) \\
&= \frac{\partial}{\partial \mu_k} \mathrm{tr}(\mu_k^2 x_k x_k^T - 2\mu_k A^T x_k x_k^T) \\
&= 2\mu_k \mathrm{tr}(x_k x_k^T) - 2\mathrm{tr}(A^T x_k x_k^T) = 0,
\end{aligned} \tag{3.15}
$$

and the final solution is

$$\mu_k = \frac{\mathrm{tr}(A^T x_k x_k^T)}{\mathrm{tr}(x_k x_k^T)} = \frac{\mathrm{tr}(x_k x_k^T A)}{x_k^T x_k} = \mathrm{tr}(x_k x_k^T A). \tag{3.16}$$

It is worth noting that $A$ is the adjacency matrix of layer $\alpha(A^{[\alpha]})$, while $x_k$ is the eigenvector of adjacency matrix of layer $\beta(A^{[\beta]})$. To verify that the final solution is a minimum, the Hessian matrix should be positive definitive. Calculating the second derivative

$$\frac{\partial^2 Z(\mu)}{\partial \mu^2} = \left[\frac{\partial^2 Z(\mu)}{\partial \mu_k \partial \mu_l}\right]_{kl}, \tag{3.17}$$

then the elements of Hessian matrix in equation (3.17) can be written as

$$\frac{\partial^2 Z(\mu)}{\partial \mu_k \partial \mu_l} = \frac{\partial}{\partial \mu_l}\left(\frac{\partial Z(\mu)}{\partial \mu_k}\right) = \frac{\partial}{\partial \mu_l}(2\mu_k \mathrm{tr}(x_l x_k^T) - 2\mathrm{tr}(A^T x_k x_k^T))$$
$$= \begin{cases} 0, & k \neq l \\ 2\mathrm{tr}(x_k x_k^T), & k = l \end{cases} = \begin{cases} 0, & k \neq l \\ 2, & k = l. \end{cases} \tag{3.18}$$

So the Hessian matrix equals $2l$ and the solution is global minimum.

### 3.3.1. Link prediction in multiplex networks

Consider a multiplex network $G(V, E^{[1]}, \ldots, E^{[M]} : E^{[\beta]} \subseteq V \times V, \forall \beta \in \{1, \ldots, M\})$ in which layer $\alpha \in \{1, \ldots, M\}$ (referred to as the target layer) has some missing links. All the other layers are referred to as auxiliary layers. The link prediction problem in multiplex networks can be defined as estimation of the existence likelihood of all non-observed links in the target layer based on the known multiplex network topology which comprises the observed links of target layer and the topology of auxiliary layers. Denote by $U$, the universal set containing all $|V|(|V| - 1)/2$ possible links in the target layer, where $|V|$ denotes the number of elements in set $V$. It is assumed that the missing links in target layer exist in the set $U - E^{[\alpha]}$ and the task of link prediction is to locate them.

Because the missing links are not known in real-world applications, the accuracy of link prediction algorithms should be tested by randomly dividing the observed links in target layer into two sets, (i) a training set $E_T^{[\alpha]}$ which is exposed to link prediction algorithm and (ii) a probe set $E_p^{[\alpha]}$ used for testing and from which no information is allowed for use in prediction. Clearly, the training set and probe set are disjoint and the union of them forms the set $E^{[\alpha]}$. In principle, the link prediction algorithm in a multiplex network provides an ordered list of non-observed links in target layer (i.e. $U - E_T^{[\alpha]}$) or equivalently gives each of them, say $(i, j) \in U - E_T^{[\alpha]}$, a score $S_{ij}^{[\alpha]}$ to quantify its existence likelihood. In this setting, the similarity score between nodes $i$ and $j$ in the target layer can be defined as

$$S_{ij}^{[\alpha]} = \sum_{\beta=1}^{M} \tilde{A}_{ij}^{[\alpha,\beta]}, \tag{3.19}$$

where $\tilde{A}^{[\alpha,\beta]}$ is the reconstruction of layer $\alpha$ by structural features of layer $\beta$ as defined in equation (3.8) and $\tilde{A}_{ij}^{[\alpha,\beta]}$ reflects the similarity between nodes $i$ and $j$ in the target layer from the perspective of layer $\beta$.

### 3.3.2. The special case of self-reconstruction

It is also worth exploring the special case of self-reconstruction, $\tilde{A}^{[\alpha,\alpha]}$. Keeping close to the notation of SPM [20], $\tilde{A}^{[\alpha,\alpha]}$ can be approximated by $\tilde{A}^{[\alpha,\alpha']}$ in which the auxiliary layer $\alpha'$ is a copy of target layer but a fraction $p^H$ of its links is randomly removed to constitute a perturbation set $\Delta E^{[\alpha]}$. So, the set of remainder links in the auxiliary layer is $E_T^{[\alpha]} - \Delta E^{[\alpha]}$. Denote by $\Delta A^{[\alpha]}$ and $A^{[\alpha']}$, the adjacency matrix related to perturbation set and the set of remainder links, respectively. Obviously, $A^{[\alpha']} + \Delta A^{[\alpha]}$ and $A^{[\alpha']}$ are the adjacency matrices of target layer and auxiliary layer, respectively. Then, the self-reconstruction problem can be expanded by equation (3.8) as

$$\tilde{A}^{[\alpha,\alpha']} = \sum_{k=1}^{N} \mu_k x_k^{[\alpha']} x_k^{[\alpha']^T}, \tag{3.20}$$

where $x_k^{[\alpha']}$ is an eigenvector of $A^{[\alpha']}$. Then, $\mu_k$ can be calculated by equation (3.16) as

$$\begin{aligned}
\mu_k &= \mathrm{tr}(x_k^{[\alpha']}x_k^{[\alpha']^T}A^{[\alpha]}) \\
&= \mathrm{tr}(x_k^{[\alpha']}x_k^{[\alpha']^T}(A^{[\alpha']}+\Delta A^{[\alpha]})),
\end{aligned} \tag{3.21}$$

and substituting the diagonalized $A^{[\alpha']}$ as

$$A^{[\alpha']} = \sum_{k=1}^{N} \lambda_k^{[\alpha']}x_k^{[\alpha']}x_k^{[\alpha']^T}, \tag{3.22}$$

in equation (3.21) and considering the orthogonality and normality of eigenvectors leads to

$$\begin{aligned}
\mu_k &= \mathrm{tr}((\lambda_k^{[\alpha']}x_k^{[\alpha']^T}x_k^{[\alpha']})(x_k^{[\alpha']}x_k^{[\alpha']^T}) + x_k^{[\alpha']}x_k^{[\alpha']^T}\Delta A^{[\alpha]}) \\
&= \lambda_k + \mathrm{tr}(x_k^{[\alpha']^T}\Delta A^{[\alpha]}x_k^{[\alpha']}),
\end{aligned} \tag{3.23}$$

which is the corrected eigenvalues as mentioned in SPM (refer to §5.5 for more details.). The statistical fluctuations due to randomness of perturbation set can be cancelled by averaging on several implementations of $\alpha'$ layer which leads to $<\tilde{A}_{ij}^{[\alpha,\alpha']}>$ that reproduces the results of SPM. This means that similarity measure introduced in equation (3.19) can be slightly modified as

$$S_{ij}^{[\alpha]} = \langle \tilde{A}_{ij}^{[\alpha,\alpha']}\rangle + \sum_{\beta=1,\beta\neq\alpha}^{M} \tilde{A}_{ij}^{[\alpha,\beta]}, \tag{3.24}$$

to incorporate SPM as the special case of self-reconstruction. This similarity measure can be directly applied to the problem of link prediction in multiplex networks and in the rest of paper is referred to as LRM. Using LRM for a target layer in a multiplex network means that all other layers are considered as auxiliary layers unless specified.

Also, two more modifications are applicable to LRM method. Using the perturbation idea of SPM, it is not necessary to show the whole target layer to LRM method at once. Instead, in each iteration, we randomly select 90% of train links in the target layer for calculation of LRM. Then, we report the average of results for 10 iterations. We refer to this method as perturbed LRM. In other words, perturbed LRM employs the eigenvectors of layer $\beta$ for reconstruction of layer $\alpha'$. Then, we have

$$S_{ij}^{[\alpha]} = \langle \tilde{A}_{ij}^{[\alpha,\alpha']}\rangle + \sum_{\beta=1,\beta\neq\alpha}^{M} \langle \tilde{A}_{ij}^{[\alpha',\beta]}\rangle, \tag{3.25}$$

as the formula for perturbed LRM.

Also, using the ideas of [35], it is possible to aggregate the information provided by different layers more efficiently. They propose two measures of layer relevance, namely, global overlap rate (GOR) and Pearson correlation coefficient (PCC). Here, we use GOR measure which is twice the ratio of the number of shared links to the total number of links in the two layers. Applying this measure in equation (3.25), the final scoring will be

$$S_{ij}^{[\alpha]} = \langle \tilde{A}_{ij}^{[\alpha,\alpha']}\rangle + \sum_{\beta=1,\beta\neq\alpha}^{M} \mu_{\alpha,\beta}^{\mathrm{GOR}}\langle \tilde{A}_{ij}^{[\alpha',\beta]}\rangle, \tag{3.26}$$

in which $\mu_{\alpha,\beta}^{\mathrm{GOR}}$ is the relevance of respective layers based on GOR measure. We refer to this method as perturbed LRM using GOR.

## 3.4. Performance evaluation

To characterize the behaviour of LRM, a comprehensive evaluation is done on Air/Train multiplex network. Figure 2 consolidates the results of this evaluation. Both Air and Train layers are considered as target layer in figure 2a,b, respectively. In addition, top rows of figures 2a,b show the results when just the leading eigenvector (corresponding to algebraically largest eigenvalue) is used and the bottom rows are considering all eigenvectors. The results of link prediction are evaluated by AUC, precision and average precision (§5.4) in left, middle and right columns, respectively.

In each subfigure, the fraction of randomly removed links from target layer varies from 0.1 to 0.9 with 0.1 increase in each step. Each data point shows the average result and the error bar determines the range of one standard deviation from the average. The results of the left column indicate that the removal of

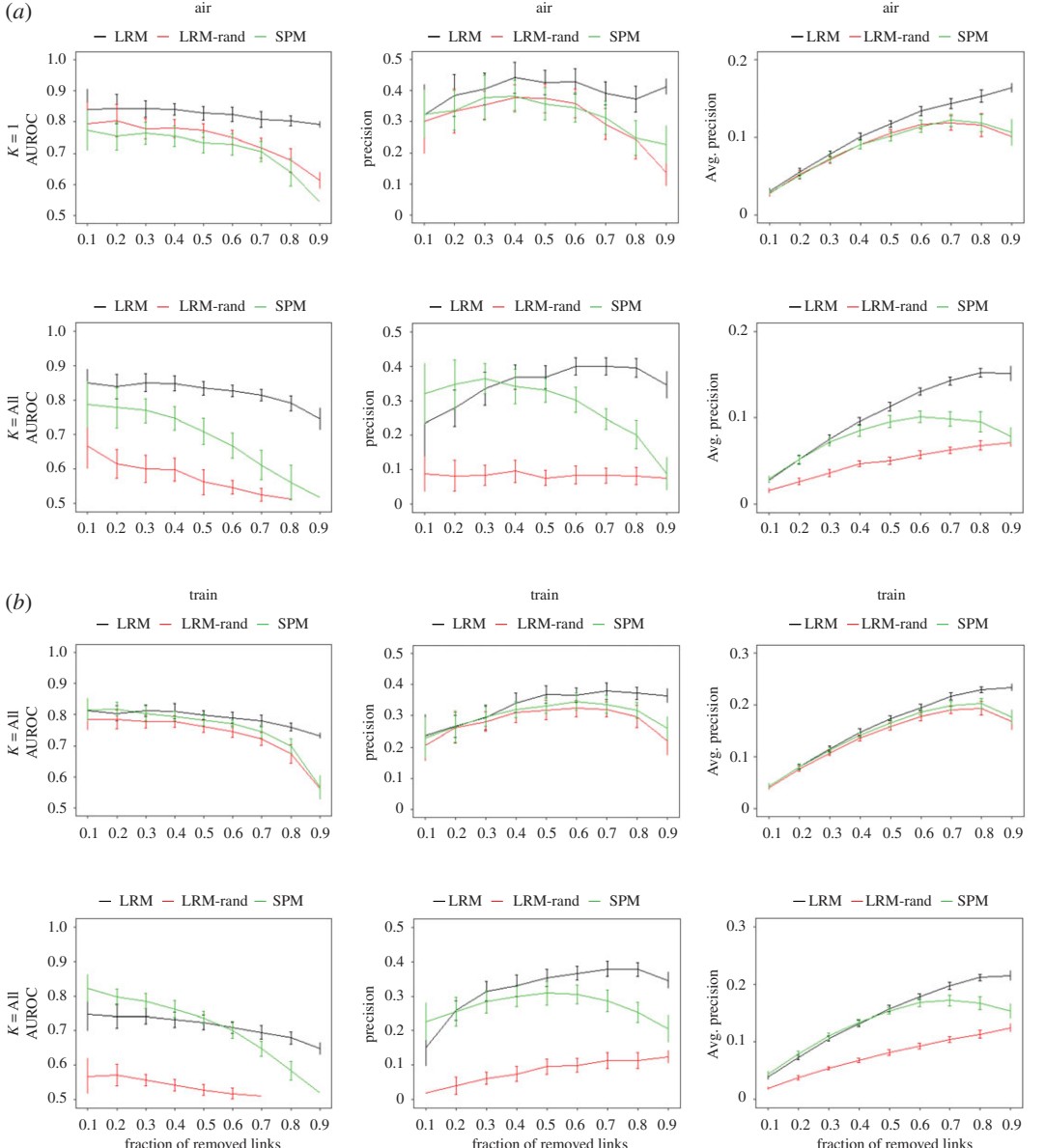

**Figure 2.** Evaluation of LRM, randomized LRM and SPM on Air/Train multiplex network. (*a*) Air layer is the target layer of link prediction and Train layer is auxiliary. (*b*) Train layer is the target layer of link prediction and air layer is auxiliary. Horizontal axes show the fraction of removed links from the target layer. Top rows in each subfigure show the results using only the leading eigenvector while the bottom rows show it using all eigenvectors. Left, middle and right columns show evaluation based on AUC (AUROC), precision and average precision measures, respectively.

more links drops the accuracy of SPM link predictor in term of AUC, which means that missing links are less likely to be scored higher than non-existent links. This result is expected as removal of more links distorts the eigenvectors as the structural features of the network. Also, the comparison of SPM results using all eigenvectors ($k$ = all) or just the leading eigenvector ($k$ = 1) does not translate to substantial change in the results, which means that the leading eigenvector contains the most important information regarding the linkage in air and train networks. Considering the fact that the leading eigenvector is related to PageRank of nodes [26] gives more insight into Air/Train network. The scores provided by PageRank in Air/Train network reflects the importance of cities from the perspective of air and train networks, respectively, while the definition of importance is based on 'the important cities are connected to other important cities'. So using SPM with the leading eigenvector only multiplies the importance of endpoint cities to assign the scores to non-observed links. In other words, it can be inferred that the missing links are likely to be found between important cities and knowing the importance of cities carries most of the information related to link prediction task in this dataset.

It is also worth noting that the results of SPM tend to resist under low fraction of missing links, specially when the leading eigenvector is used. This is due to the fact that the importance of cities is not expected to change under removal of low fraction of random links. Although, this does not hold under high fraction of removed links and this is where LRM comes to help. When the observed links of Air network do not suffice to infer an accurate importance of cities, LRM uses the importance inferred by Train network to mitigate the lack of information for link prediction. Actually, this works because these two layers are similar w.r.t. structural features, i.e. their view of importance of cities is very similar to each other. This fact can be verified again as the same holds when Train network is the target layer.

The importance of similarity of layers w.r.t. structural features can be understood more by LRM-rand, which applies LRM but uses the randomized auxiliary layer. The randomized auxiliary network is an Erdos–Renyi network with the same number of nodes and link density of original auxiliary layer. The results of the left column confirms that randomization of auxiliary layer drops the performance of LRM while destroying the similarity of two layers w.r.t. structural features. Also, it is clear that the negative effect of random auxiliary layer increases as all eigenvectors are being used. Therefore, here using the leading eigenvector leads to stable and superior results of LRM, while makes it more robust against random auxiliary network.

The middle column of figure 2 shows the results based on the precision measure. While AUC evaluates the whole list of scored non-observed links, the precision metric evaluates the top entries of the list. The number of top entries of the list which are used for evaluation is equal to the number of elements in the probe set. Therefore, as the fraction of removed links grows, the number of top entries of the list that are used for evaluation grows as well. Usually, in low fraction of missing links (say the range of [0.1–0.3]) AUC does not change significantly. This leads to higher fraction of missing links in the top entries of the list and increase of the precision metric until the AUC falls and that makes the precision fall as well. Evaluation of LRM by the precision metric confirms that using the leading eigenvector is a better option for Air/Train dataset as it gives higher performance and is more robust against random auxiliary layer. Also, it can be inferred that it is more difficult to increase the precision under low fraction of missing links. In addition, it is clear that LRM overall is able to increase the precision of link prediction in this dataset and specially avoids the fall of the performance under high fraction of removed links.

The right column of figure 2 shows the results based on average precision metric. The average precision metric considers the entries of the list from the topmost to the last missing link, determines the precision at the cut-off of each missing link and outputs the average of the precision values. So, the higher values of average precision indicate the higher concentration of missing links towards the top of the list of non-observed links. The results clearly support that LRM is able to concentrate the missing links towards the top of the list, and it does it better as the fraction of removed links grows. Once again, it can be confirmed that using the leading eigenvector is an appropriate choice and makes the results robust against random auxiliary layer.

The effectiveness of LRM is also verifiable using a synthetic network in which a backup of target layer comes to help. Figure 3 shows the results of SPM and LRM link prediction performance on a synthetic multiplex network which is made of a duplication of air layer in Air/Train network. This network is referred to as air/air-backup network. The air layer is the target layer. While a fraction of links is removed from target layer for link prediction purpose, the air-backup layer remains untapped and contains all the information about missing links. LRM is expected to benefit most out of the information of auxiliary layer. When all eigenvectors are used, the results shown in figure 3a support that LRM works perfectly (AUC = 1) as long as at least $N$ edges are known. Once again, the result for using the leading eigenvector figure 3b confirms that much of information about the linkage in Air network is contained in the leading eigenvector and LRM is able to transfer it to the target layer in an effective manner. A slight increase of performance at the high fraction of removed links is a special case for this synthetic network. Large fraction of removed links leads to smaller magnitude of scores provided by LRM(self). As the information transferred from auxiliary layer is rich, the addition of LRM(self) degrades the performance. This degradation of performance is less for smaller magnitude of LRM(self) scores, so it seems that the overall performance is increasing.

The issue of choosing the right number of leading eigenvectors can be clarified more by using results depicted in figure 4. Here, the dataset under study is brain network and the target layer of link prediction is structure layer. The results of SPM and LRM are shown versus number of leading eigenvectors used. Specifically, comparing the results reveals that just using the leading eigenvector is not enough, in contrast to Air/Train network, and the performance degrades slightly when more than 15 leading

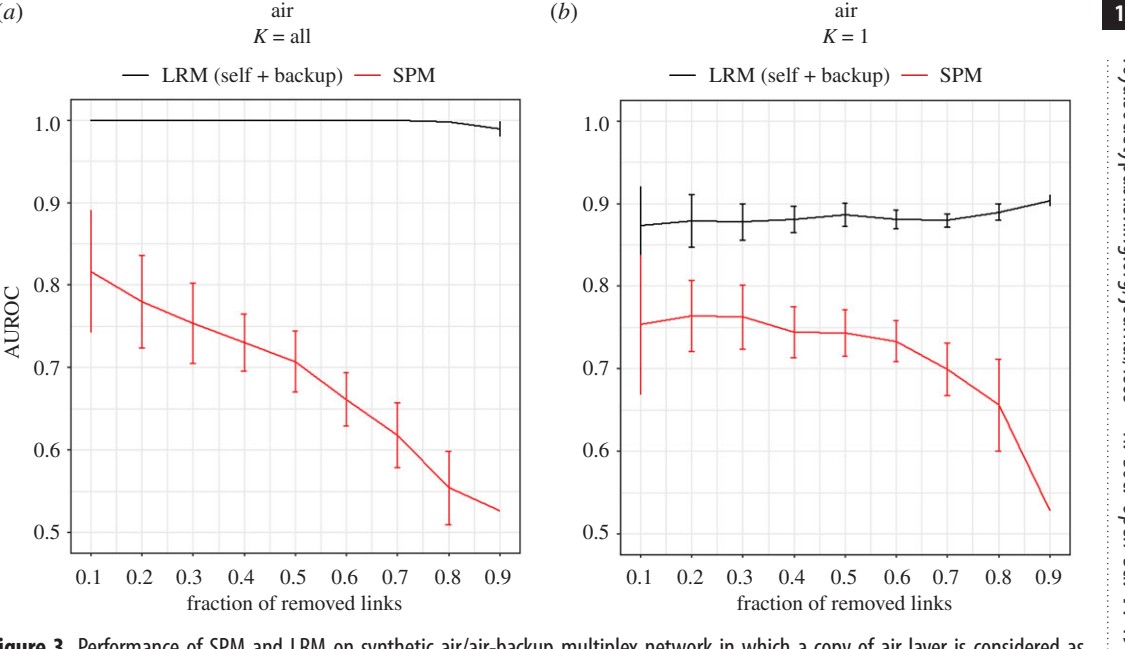

**Figure 3.** Performance of SPM and LRM on synthetic air/air-backup multiplex network in which a copy of air layer is considered as the auxiliary layer. The links are removed from the target layer while the auxiliary layer remains untapped. (*a*) Using all eigenvectors. (*b*) Using the leading eigenvector. The horizontal axis is the fraction of removed links and the vertical axis shows the performance based on AUC. The result supports that LRM is efficiently able to transfer information from the auxiliary layer to the target layer.

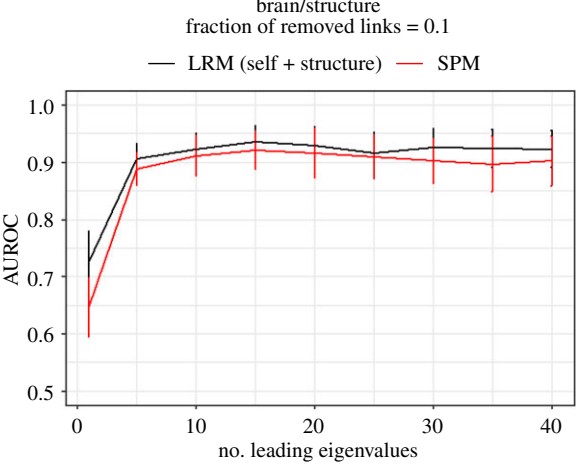

**Figure 4.** Performance of LRM and SPM for different number of leading eigenvectors. The horizontal axis shows the number of leading eigenvectors used in the methods. The vertical axis shows the performance based on AUC. The result indicates that using an appropriate number of leading eigenvectors leads to good performance while minimizing the susceptibility to noise-like patterns.

eigenvectors are used. So, choosing the top 15 leading eigenvectors for both SPM and LRM seems to be a reasonable choice, because keeping $k$ as small as possible is favourable due to increase of robustness against noise-like patterns in data and less computational complexity.

Sample results of evaluation of SPM and LRM on CS-Aarhus, *C. elegans* and Physicians multiplex networks are shown in figure 5. In CS-Aarhus as depicted in figure 5*a*, the lunch layer is considered as the target layer for link prediction. All other layers including FB, co-author, leisure and work are considered as auxiliary layers one by one. Results indicate that using each of these layers has positive impact on the performance of LRM but the work and leisure layers have the highest impact. This is compatible with the intuition that co-workers and those who go together for leisure are more likely to have lunch with each other. Also, this raises the idea of using both of these two layers in LRM, which gives a superior performance, as can be seen in figure 5*a*.

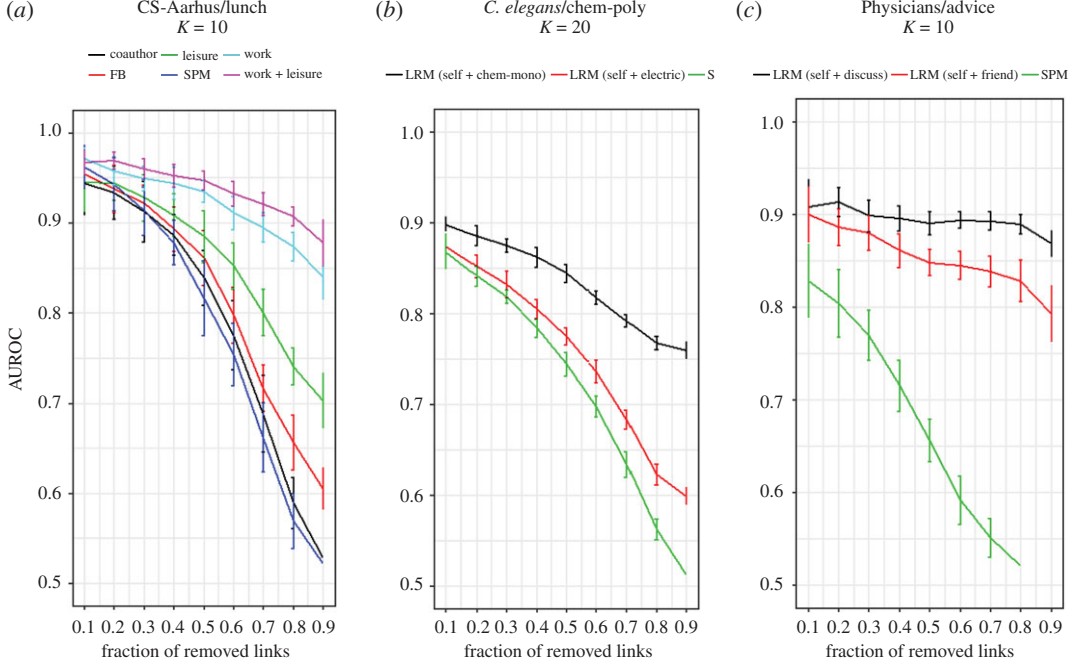

**Figure 5.** Evaluations of SPM and LRM on (*a*) CS-Aarhus, (*b*) *C. elegans* and (*c*) Physicians multiplex networks. The performance is measured using AUC for different fraction of removed links from target layer. In CS-Aarhus network, the lunch layer is considered as target layer and methods use 10 leading eigenvectors. All other layers are used as auxiliary layers. The work and leisure layers are boosting the performance of LRM most and using them at the same time leads to an outstanding result. In *C. elegans* network, the target layer is chem-poly. Using chem-mono layer of chemical nature enhances the performance more noticeably compared with electric layer. In Physicians network, the target layer is advice. Both discuss and friend layers are good at improving the performance of LRM but discuss layer shows a superior result.

The results of LRM for *C. elegans* network are also notable, as shown in figure 5*b*. Here, the target layer is chem-poly which is of chemical nature. Although using the electric layer as auxiliary layer improves the performance of LRM, using chem-mono layer, which is of the same nature as the target layer, has much more positive impact on the performance of link prediction. This result supports the core idea that layers which have similar organizations are more helpful for link prediction process.

The performance of SPM and LRM on Physicians is shown in figure 5*c* when advice layer is the target layer for link prediction. The results suggest that both discuss and friend layers are very helpful for predicting missing links in advice layer. Furthermore, the result of LRM using discuss layer indicates that when almost nothing is left from advice layer, information from discuss layer gives the ability to discriminate between missing links and non-existent links in advice layer.

The performance of SPM on Tube layer of LondonTransport is shown in figure 6*a*. The poor performance of SPM is an indicator of low link predictability of this network [20]. Also, the results of LRM show that other layers do not help to overcome this problem. In addition, comparing the results of LRM with LRM-rand, in which the auxiliary layer is randomized, supports that overground and DLR layers seem like random layers to Tube layer. This is also consistent with the results seen in table 1, which do not consider the similarity of structural features of these layers as statistically significant. Therefore, it can be concluded that LondonTransport network is a hard case for link prediction task, both in simplex and multiplex settings.

The performance of SPM on suppress layer of *Drosophila* network is shown in figure 6*b*, and the result is quite similar to other datasets excluding LondonTransport. What makes this dataset distinct from others is that LRM does not improve the performance. The result for *k* = 30 is shown in figure 6*b* but several tries with upper and lower values did not lead to remarkable change. Looking into table 1 reveals that, although the value of similarity of structural features between two layers of *Drosophila* network is statistically significant, the value is lowest among all datasets. Therefore, it can be concluded that statistical significance of similarity of structural features is a necessary condition for LRM to work, but not the sufficient condition, and a practically significant similarity is needed for LRM to work in practice.

**Figure 6.** Evaluation of SPM and LRM on (*a*) LondonTransport and (*b*) *Drosophila* multiplex networks. These two networks are examples of hard cases of link prediction task. The target layer in LondonTransport network is the Tube layer. Low performance of SPM indicates the hardship of link prediction in this layer. As the LRM performance is low for both overground and DLR as auxiliary layers, it can be inferred that the link prediction is hard in multiplex setting as well. In *Drosophila* network, the suppress layer looks predictable but the results of LRM indicates that it is hard to extract information from additive layer to help link prediction in suppress layer.

## 3.5. Comparison with the state-of-the-art methods

In this section, we compare the results of our methods against the state-of-the-art methods. The fraction of removed links from the target layer is set to 0.1 for all experiments. The application of SPM on target layer, SPM(T), and simple addition of SPM scores for target and auxiliary layers, SPM(T+A), are considered as a baseline methods. In [35], authors apply well-known single-layer similarity measures like common neighbours (CN), resource allocation (RA) and local path index (LPI) to each layer of a multiplex network as baseline measure and then combine the produced scores based on a tunable parameter $\phi$ and a layer relevance measure. Two layer relevance measures namely, global overlap rate (GOR) and Pearson correlation coefficient (PCC) are used. The authors declare that for a wide range of datasets $\phi = 0.5$ is an appropriate choice. We applied their method with different baseline measures but we report the result of the best performing baseline measure for succinctness. The results for both GOR and PCC considering $\phi = 0.5$ are reported in table 3. In [34], authors have proposed three single-layer baseline measures, namely, WCN, HP and Rank-CN-HP. WCN uses embedded network in geometric space and calculates hyperbolic distance of nodes to weigh the importance of common neighbours. HP considers the hyperbolic distance of nodes as a dissimilarity measure. Rank-CN-HP uses the rank of nodes pairs instead of their score based on CN and HP. Similar to [35], they combine the score or ranks of each layer using GOR and $\phi = 0.5$. Our proposed methods are LRM (equation (3.19)), perturbed LRM (equation (3.25)) and perturbed LRM with GOR (equation (3.26)).

The link prediction methods in single-layer networks have been frequently compared with each other based on their performance on real networks while much less is done for multiplex networks. The results provided in table 3 are a step for alleviating this shortcoming. In almost all cases, the only single-layer method, SPM(T), is not able to compete with the multiplex methods. This confirms the known fact that in real-world multiplex networks, layers are informative for link prediction tasks of each other. Although among multiplex link prediction methods no-one is winner for all cases, our proposed LRM_PG outperforms others both in total number of best results (12/29 while the second best method YaoPL achieves 5/29) and in pairwise comparison (LRM_PG: 18/29 cases versus the second best method, YaoGL: 11/29 cases). This can be attributed to the special way of information transfer from the auxiliary to the target layer in which both layers are involved and is unique to our proposed methods.

**Table 3.** Performance evaluation of the link prediction methods on 29 real-world duplex networks based on AUC measure. Three left columns determine the name of the multiplex networks, the target layer of link prediction, and the auxiliary layer which comes to help the prediction task. From left to right, the evaluated methods are SPM(T): a single-layer method that applies SPM on target layer as a baseline, LRM: our proposed layer reconstruction method, LRM_P: our proposed perturbed LRM, LRM_PG: our proposed perturbed LRM using GOR as the layer relevance measure, SPM(T + A): addition of scores provided by applying SPM on both the target and the auxiliary layers, YaoPL: a state-of-the-art method that uses LPI with PCC as the layer relevance measure, YaoGL: LPI with PCC as the layer relevance measure, SamieHP: a state-of-the-art method that uses hyperbolic distance as dissimilarity measure within each layer with GOR as the layer relevance measure. The best result in each row is shown in bold.

| dataset name | target layer | auxiliary layer | SPM(T) | LRM | LRM_P | LRM_PG | SPM(T + A) | YaoPL | YaoGL | SameiHP |
|---|---|---|---|---|---|---|---|---|---|---|
| Air/Train | air | train | 82.58 | 87.93 | 87.60 | **90.32** | 89.54 | 86.74 | 86.29 | 88.21 |
| | train | air | 81.21 | 82.72 | **85.05** | 84.84 | 82.24 | 82.61 | 84.11 | 79.23 |
| **C. elegans** | electric | chem-mono | 78.66 | 82.84 | 84.17 | 83.12 | 82.84 | **84.20** | 83.66 | 75.32 |
| | | chem-poly | 76.85 | 83.91 | 85.23 | **86.62** | 85.97 | 85.59 | 84.39 | 78.23 |
| | chem-mono | electric | 81.04 | 80.70 | 81.85 | 82.58 | 81.93 | 83.63 | **84.09** | 73.65 |
| | | chem-poly | 80.57 | 91.33 | **91.94** | 91.89 | 91.54 | 87.94 | 88.69 | 74.52 |
| | chem-Poly | electric | 86.86 | 85.87 | 86.80 | **87.20** | 87.01 | 86.83 | 86.44 | 70.44 |
| | | chem-mono | 86.07 | 90.09 | 89.54 | **90.13** | 89.79 | 87.93 | 87.45 | 70.53 |
| human brain | structure | function | 90.63 | 90.98 | 90.84 | **93.91** | 91.47 | 93.17 | 93.58 | 68.76 |
| | function | structure | 88.38 | 89.93 | 89.60 | 90.87 | 88.77 | 90.85 | **91.60** | 78.17 |
| Physicians | advice | discuss | 81.19 | 90.06 | 91.12 | 91.68 | 90.99 | 92.27 | **92.81** | 82.79 |
| | | friendship | 84.93 | **89.79** | 89.32 | 89.12 | 89.79 | 89.10 | 88.53 | 81.92 |
| | discuss | advice | 80.83 | 89.18 | 90.86 | 90.26 | 89.92 | 90.61 | **91.98** | 79.74 |
| | | friendship | 83.37 | 89.29 | 90.19 | 89.46 | **90.90** | 90.06 | 90.34 | 80.67 |
| | friendship | advice | 84.69 | **88.79** | 87.67 | 86.94 | 87.76 | 86.49 | 86.98 | 78.46 |
| | | discuss | 79.23 | 89.02 | 89.84 | 89.24 | **90.93** | 90.06 | 89.88 | 81.08 |
| CS-Aarhus | lunch | Facebook | **95.94** | 93.16 | 92.55 | 95.87 | 92.68 | 95.01 | 93.38 | 80.36 |
| | | co-author | 95.86 | 93.62 | 94.98 | **96.01** | 94.91 | 94.50 | 94.37 | 74.59 |
| | | leisure | 96.11 | 95.53 | 94.71 | **96.23** | 94.39 | 93.96 | 94.16 | 77.19 |
| | | work | 94.54 | 94.95 | 95.10 | **96.79** | 95.04 | 94.58 | 95.60 | 78.07 |

(Continued.)

**Table 3.** (*Continued.*)

| dataset name | target layer | auxiliary layer | SPM(T) | LRM | LRM_P | LRM_PG | SPM(T + A) | YaoPL | YaoGL | SameiHP |
|---|---|---|---|---|---|---|---|---|---|---|
| CS-Aarhus | Facebook | lunch | 91.32 | 87.78 | 87.66 | 92.75 | 87.42 | **94.88** | 94.63 | 78.88 |
| | | co-author | 92.16 | 91.78 | 92.93 | 91.57 | 91.39 | **94.89** | 94.09 | 75.68 |
| | co-author | lunch | 66.64 | 92.73 | 93.28 | **95.85** | 91.58 | 92.01 | 91.71 | 76.38 |
| | | Facebook | 81.67 | 80.37 | 80.95 | **87.90** | 78.99 | 78.95 | 76.09 | 77.98 |
| | leisure | lunch | 76.49 | **91.26** | 90.30 | 91.06 | 90.53 | 89.85 | 86.78 | 80.96 |
| | work | lunch | 63.67 | 62.63 | 64.78 | **91.10** | 83.28 | 89.67 | 90.41 | 80.76 |
| | | Facebook | 63.00 | 91.90 | 90.28 | 89.05 | 91.61 | **95.50** | 86.09 | 78.49 |
| | | co-author | 63.94 | 83.86 | 82.43 | **88.61** | 84.34 | 85.52 | 86.76 | 77.40 |
| | | leisure | 64.18 | 87.65 | 89.40 | 90.00 | 88.73 | **93.54** | 86.19 | 79.21 |

# 4. Discussion

Studies of multiplex networks are a major step towards understanding of real-world complexity. These studies should clarify how different layers interact to shape the function of each layer and the function of the system as a whole. This needs an in-depth understanding of multiplex network structure. The study of link prediction is a key to unfold the structural features of multiplex networks.

Our study provides evidence about similarity of structural features of layers in real social, biological and technological multiplex networks. Also, it has shown that similarity w.r.t. structural features is a major source of information redundancy, and LRM is able to use it to enhance the performance of link prediction in these networks. In addition, results of experiments support that using a sufficient number of leading eigenvectors yields the desired performance and makes the method robust against noise-like structural features.

It is notable that LRM considers the issue of correlated multiplexity. Rotation of one layer neither changes the eigenvalues nor the corresponding eigenvectors. The value assigned to each node by each eigenvector does not change. On the other hand, representing these values as a vector needs an order of nodes and this is where rotation plays its role. Therefore, rotation of a layer permutes the elements of the eigenvectors and destroys the similarity of layers w.r.t. structural features. Obviously, this leads to loss of performance in LRM. Also, it can be inferred that if, in a real multiplex network, the multiplexity of layers is unknown at first, then an initial phase of network alignment [45] is helpful for finding the right one-to-one mapping of nodes. Then, LRM works if layers show enough similarity w.r.t. structural features.

Finally, it is possible to sort the links (including observed and non-observed links) in target layer only according to the part of LRM score which comes from the auxiliary layer. In this way, the links in target layer which are more compatible with the structural features of auxiliary layer will be on the top of the list. For example, in CS-Aarhus network the links in lunch layer can be sorted both according to LRM(work) and LRM(leisure) and the top entries of the lists will be the links which are more compatible with the structural features of work and leisure layers, respectively.

The link prediction problem has attracted increasing attention from both physical and computer science communities because of its broad applications [21]. In biological networks such as protein–protein interactions, the discovery of links is costly and the cost increases when multiple types of links are involved. This study showed that if some types of interactions are better known in these networks, the discovery of few links of less-known interactions facilitate the discovery of the rest of missing links. In social networks, there are numerous contexts of relationship among human beings that many of them are less investigated. The human network is not well-understood unless these contexts of relations are taken into consideration. For example, human beings refer to each other for different affairs like education, healthcare and business. Some of these relations are more disclosed and some others are less disclosed, and the only way to acquire more information about the latter is by leveraging the information contained in former. This is where methods like LRM come to help. The value of similarity w.r.t. structural features indicates which known relations should come to help. In technological networks like air transportation systems, a very tough competition exists among different airlines. Here, it is always an advantage for an airline to know which new airways the rival company will run in future. It can be said that the answer comes not only from the network of rival company but also from the networks of airlines similar to that rival.

# 5. Methods

## 5.1. Data introduction

The real-world multiplex datasets which are under study in this work can be categorized as social (Physicians, CS-Aarhus), biological (Brain, *C. elegans*, *Drosophila*) and technological (Air/Train, LondonTransport). An overview of the datasets and related statistics can be found in table 2. A brief explanation about datasets is given below

*Physicians*. This dataset is about three types of relations among US physicians in four towns [46]. Layers correspond to advice, discussion (abbreviated as discuss) and friendship (abbreviated as friend) relations among the physicians, respectively.

*CS-Aarhus*. This multiplex social network consists of five types of online and offline relationships between the employees of Computer Science department at Aarhus University, Aarhus, Denmark [44].

Layers correspond to relationship via lunch, Facebook (abbreviated as FB), co-authorship (abbreviated as co-author), leisure and work.

*Brain*. Two modes of connectivity between regions of human brain is covered in this dataset [47]. One mode consists of structural (abbreviated as structure) network among brain regions and is obtained by setting a threshold on connection probability between brain region pairs measured using diffusion magnetic resonance imaging (dMRI) [28]. The other mode is the functional (abbreviated as function) network of brain regions which is derived by setting a threshold on correlation of activities of brain region pairs and is measured using blood oxygen level-dependent functional magnetic resonance imaging (BOLD fMRI) [28].

*Caenorhabditis elegans*. Three types of synaptic connections among neurons of the nematode *Caenorhabditis elegans* are characterized in this dataset [48]. These connections are electrical (abbreviated as electric), chemical monadic (abbreviated as chem-mono) and chemical polyadic (abbreviated as chem-poly).

*Drosophila*. The *Drosophila melanogaster* is a species of fly and is also known generally as common fruit fly [49]. The dataset represents two types of genetic interaction among proteins of this insect. One layer corresponds to suppressive genetic interaction (abbreviated as suppress), while the other corresponds to additive genetic interaction (abbreviated as additive).

*Air/Train*. This dataset contains air and train transportation network of India [28,50]. Each node of the network represents a supernode that contains an airport and train stations within 50 km from that airport. Obviously, supernodes are connected through flights to and from the air network. In the train network, two supernodes are connected if they share a train station or if they are directly connected to a train station.

*LondonTransport*: These data were collected in 2013 from the official website of Transport for London [51]. Nodes are train stations in London including underground, overground and DLR stations. Layers correspond to connectivity of stations via underground line (known as Tube), overground line or DLR, respectively.

## 5.2. Data statistics

Table 2 shows the major statistics of multiplex networks under study. The number of nodes in each multiplex network equals the number of nodes which are active in at least one layer while the node multiplexity is the fraction of nodes which are active in more than one layer. The number of active nodes in each layer equals the number of nodes which have at least one link in that specific layer and may differ from the number of nodes in multiplex network.

## 5.3. Link prediction problem

The link prediction problem arises in the networks in which some of the links are missing or may be added in future. The link prediction algorithms are supposed to estimate the existence likelihood of all non-observed links based on the observed links of the network. Consider a simple network $G(V, E)$ in which $V$ and $E$ are sets of nodes and links. Denote by $U$ the universal set of all possible $(|V| \times |V-1|)/2$ links in the network, where $|V|$ is the number of elements in set $V$. So, the set $U - E$ will be the set of non-observed links of the network that contains the missing links which link prediction algorithms are supposed to locate them.

As the missing links are not known beforehand in real applications, to investigate the suitability of link prediction algorithms, some of the links of network should be removed randomly to form the probe set $E_p$ and the remainder links are considered as training set $E_T$. Obviously, $E_T \cup E_p = E$ and $E_T \cap E_p = \emptyset$. The link prediction algorithms are allowed to use $E_T$ to locate $E_p$ among all possible choices in $U - E_T$ and if they do well, hopefully they can do the same for missing links in $U - E$ for which there is no ground truth. For this purpose, the link prediction algorithms assign an existence likelihood score $S_{ij}$ to each non-observed link $(i, j) \in U - E_T$.

## 5.4. Evaluation metrics

The link prediction algorithms provide an existence likelihood score for each non-observed link which can be used for sorting them from more likely to less likely missing links. A perfect sorting put the missing links at the top of the list and all other non-existent links underneath. To measure how far the sorted list by a link prediction algorithm is from the perfect sorting, some evaluation metrics are needed. Three standard evaluation metrics are area under the receiver operating characteristic curve

(AUC or AUROC) [52], precision [53] and average precision. The first measure evaluates the whole list and the other two evaluate the top of the list.

*AUC*. This measure shows the probability that a randomly chosen missing link has higher score than a randomly chosen non-existent link. A good estimate of this measure can be achieved by sampling. A random sample from each of missing links and non-existent links is picked at each time. Considering $n$ independent samples out of which $n'$ times, the missing link has higher score than the non-existent link and $n''$ times they have the same score. Then, the AUC can be calculated as

$$\text{AUC} = \frac{n' + 0.5 \times n''}{n}. \tag{5.1}$$

Random assignment of scores leads to the AUC value of approximately 0.5. As the sorting gets closer to perfect sorting, the value of AUC approaches 1. In this way, the AUC measure evaluates the quality of the whole list.

*Precision*. The sorted list of non-observed links is expected to put the missing links at the top of the list. Considering $L = |E_p|$ as the total number of missing links, the top $L$ entries of the sorted list can be examined to see whether they are missing links (denoted by $L_r$) or non-existent links. Precision value can be calculated as

$$\text{precision} = \frac{L_r}{L}. \tag{5.2}$$

*Average precision*. The precision@k is the value of precision for top $k$ entries of the sorted list of non-observed links. So it is possible to calculate the precision from the top entry of list to each missing link in the list. So for each missing link, there will be a precision value and the average of these values is the average precision.

## 5.5. Structural perturbation method

The SPM is based on a fundamental hypothesis that missing links are difficult to predict if their addition causes huge structural changes and thus, a network is highly predictable if the removal or addition of a set of randomly selected links does not significantly change the network's structural features (i.e. eigenvectors) [20]. So, the missing links of a network if added are supposed to just change the eigenvalues but not the eigenvectors of the network. This adjustment to eigenvalues can be calculated by removing a set of randomly selected links which are known as perturbation set. The fact that independent perturbation sets lead to correlated adjustment values means that a generalization is happening and gives SPM the capability to predict the missing links. Applying SPM for link prediction and evaluation of results is done according to following steps:

1. Divide observed network $A$ into training set $E_T$ and probe set $E_p$, obviously, $A = A_T + A_p$.
2. Furthermore, randomly divide the set $E_T$ into remainder set $E_R$ and perturbation set $\Delta E$ and denote their adjacency matrices as $A_R$ and $\Delta A$, respectively.
3. Calculate the eigenvalues $\lambda_k$ and their corresponding eigenvectors $x_k$ of $A_R$.
4. Calculate $\Delta \lambda_k = x_k^T \Delta A x_k$.
5. Calculate the perturbed matrix $\tilde{A} = \sum_{k=1}^{N} (\lambda_k + \Delta \lambda_k) x_k x_k^T$.
6. Repeat steps 2 to 5 ten times and use the average of ten $\tilde{A}$, denoted by $< \tilde{A} >$, as the final score, where $\langle \tilde{A} \rangle_{ij}$ is the score of link $(i, j)$.
7. Evaluate the scores of non-observed links (i.e. links in $U - E_T$ where $U$ is universal set of all possible links) by AUC or precision (as mentioned in evaluation metrics).
8. Repeat steps 1 to 8 $n$ times (in this paper $n = 30$) and report the average of AUC or Precision.

Data accessibility. The datasets used in this study are the property of their respective owners. All datasets are available for free and their respective authors are cited properly in §5.1. Using Air/Train dataset needs a permission from authors that we obtained. The relevant code for this research work are stored in GitHub: https://github.com/UT-NSG/LRM and have been archived within the Zenodo repository: https://doi.org/10.5281/zenodo.3760786 The code developments are built upon this package: https://github.com/galanisl/LinkPrediction
Authors' contributions. A.M.A.-Q., M.A., N.Y. and M.R. designed the research question. A.M.A.-Q. and S.H.J. conceived the method idea. A.M.A.-Q. led the study. A.M.A.-Q., S.H.J. and A.T. analysed the data. A.M.A.-Q. and A.T. did the mathematical modelling. A.T. and S.H.J. performed the coding. A.M.A.-Q. and A.T. designed experiments. A.M.A.-Q.,

S.H.J. and A.T. conducted the experiments. A.M.A.-Q. wrote the paper. N.Y. and M.R. did proof-reading and commenting. N.Y., M.A. and M.R. supervised the research.

Competing interests. The authors declare no competing interests.

Funding. The authors received no specific funding for this work.

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
