## [Reviewer comments · Royal Society Open Science]

Review History

RSOS-191928.R0 (Original submission)

Review form: Reviewer 1

Is the manuscript scientifically sound in its present form?

Yes

Are the interpretations and conclusions justified by the results?

Yes

Is the language acceptable?

No

Do you have any ethical concerns with this paper?

No

Have you any concerns about statistical analyses in this paper?

No

Recommendation?

Major revision is needed (please make suggestions in comments)

Comments to the Author(s)

This manuscript has proposed a new link prediction method for multiplex networks. The results are interesting and I do support the acceptance provided that the authors address the following comments:

- There is mix of concepts in the abstract. While the authors refer to heterogeneity of entities, they immediately mention multiplex networks. In multiplex networks, all the nodes are from the same type, but link could be developed in different layers. I suggest removing the first.
- Writing needs significant improves and the manuscript should go under a careful proofreading. The fonts on the math within text is much larger than the the rest of the text, which make it difficult to follow them. My advice is to convert everything to LaTeX.
- Eigenvectors of an adjacency matrix contain only some of its structural information, e.g. one can partition the network based on the eigenvector corresponding to the largest eigenvalue.
- I am still not clear why similarity between the eigenvectors should be meaningful; detailed discussion is required for that. They authors may discuss relevance of these similarities to the existing literature, e.g. doi.org/10.1098/rsos.171747.
- Figure captions should be extended by including further details. This will make them more readable.
- The results have been compared with only a single baseline method, which is a rather old publication (2015). The authors should compare the performance with more state-of-the methods, e.g. doi.org/10.1093/comnet/cnz007. I also noticed that some of the references cited in the manuscript have also proposed link prediction methods for multiplex networks. They also need to be compared with.

Review form: Reviewer 2

Is the manuscript scientifically sound in its present form?

Yes

Are the interpretations and conclusions justified by the results?

Yes

Is the language acceptable?

Yes

Do you have any ethical concerns with this paper?

No

Have you any concerns about statistical analyses in this paper?

No

Recommendation?

Accept with minor revision (please list in comments)

Comments to the Author(s)

The work has proposed an approach for link prediction in Multiplex Networks and it has illustrated the benefit of applying multiple layers in link prediction compared to using just a single layer. The paper may be accepted subject to appropriate revision addressing the following comments.

Major Remark

=====

1- Making a statement about causality based on observing correlation or similarity is not valid, while the reverse is valid. Therefore, on page 6, Lines 14-17, the statement "In other words, the major driving forces behind the organization of these networks are similar. Made simple by example, if the triadic closure is a driving force behind the organization of Advice network then it is very likely that the same holds for Discuss layer." is not valid. This statement seems intuitively valid, but scientifically is wrong. According to the context of the paper, it is recommended that limit the conclusions in the level of the similarity and do not generalize into the concept of causality.

Minor Remarks

=====

2- In Figure 3, for $K=1$ (Left figure), it can be observed overall that AUROC for LRM increases with the fraction of removed links. Please explain this observation (as we expect intuitively to see a reduction in AUROC by increasing the fraction of removed links)?

3- In Fig. 4, we see three figures corresponding to three different values of K and we still do not know the optimum value of K and we can see just the trend of the changes. It is recommended to have just one figure showing AUROC for a fixed fraction of removed links vs. different values of K . We can have multiple curves for different values of the fraction of removed links.

Grammar & Typos

=====

4-Please read the paper to correct some grammar mistakes.

5- Punctuations need to be revised (e.g., when you have "... A,B, and C..." you need comma before "and". For example, on page 2 lines 36-38 you have missed comma before "and": Real world networks like biochemical, human (HERE) and air transportation networks are examples of biological, social (HERE) and technological systems, respectively.). In many places, commas are missed. For example, on page 2 line 46, "In a similar way, (YOU NEED COMMA HERE) people are ...".

6- Page 6, Line 50: "The results of Table 1 indicates that" -> "The results of Table 1 indicate that"

7- Page 13, Line 30: "...can be clarified more using results in Figure 4." -> "can be clarified more by using the results depicted in Figure 4."

Decision letter (RSOS-191928.R0)

10-Mar-2020

Dear Mr Abdolhosseini-Qomi,

The editors assigned to your paper ("Link Prediction in Real-World Multiplex Networks via Layer Reconstruction Method") have now received comments from reviewers. We would like you to revise your paper in accordance with the referee and Associate Editor suggestions which can be found below (not including confidential reports to the Editor). Please note this decision does not guarantee eventual acceptance.

Please submit a copy of your revised paper before 02-Apr-2020. Please note that the revision deadline will expire at 00.00am on this date. If we do not hear from you within this time then it

will be assumed that the paper has been withdrawn. In exceptional circumstances, extensions may be possible if agreed with the Editorial Office in advance. We do not allow multiple rounds of revision so we urge you to make every effort to fully address all of the comments at this stage. If deemed necessary by the Editors, your manuscript will be sent back to one or more of the original reviewers for assessment. If the original reviewers are not available, we may invite new reviewers.

- Data accessibility

If you wish to submit your supporting data or code to Dryad (<http://datadryad.org/>), or modify your current submission to dryad, please use the following link:
<http://datadryad.org/submit?journalID=RSOS&manu=RSOS-191928>

- Competing interests

- Authors' contributions

- Acknowledgements

- Funding statement

Kind regards,

Andrew Dunn

on behalf of Professor Matjaz Perc (Associate Editor) and Miles Padgett (Subject Editor)

Comments to Author:

Reviewers' Comments to Author:

Reviewer: 1

Comments to the Author(s)

This manuscript has proposed a new link prediction method for multiplex networks. The results are interesting and I do support the acceptance provided that the authors address the following comments:

- There is mix of concepts in the abstract. While the authors refer to heterogeneity of entities, they immediately mention multiplex networks. In multiplex networks, all the nodes are from the same type, but link could be developed in different layers. I suggest removing the first.
- Writing needs significant improves and the manuscript should go under a careful proofreading. The fonts on the math within text is much larger than the rest of the text, which make it difficult to follow them. My advice is to convert everything to LaTeX.
- Eigenvectors of an adjacency matrix contain only some of its structural information, e.g. one can partition the network based on the eigenvector corresponding to the largest eigenvalue.
- I am still not clear why similarity between the eigenvectors should be meaningful; detailed discussion is required for that. They authors may discuss relevance of these similarities to the existing literature, e.g. doi.org/10.1098/rsos.171747.
- Figure captions should be extended by including further details. This will make them more readable.
- The results have been compared with only a single baseline method, which is a rather old publication (2015). The authors should compare the performance with more state-of-the methods, e.g. doi.org/10.1093/comnet/cnz007. I also noticed that some of the references cited in the

manuscript have also proposed link prediction methods for multiplex networks. They also need to be compared with.

Reviewer: 2

Comments to the Author(s)

The work has proposed an approach for link prediction in Multiplex Networks and it has illustrated the benefit of applying multiple layers in link prediction compared to using just a single layer. The paper may be accepted subject to appropriate revision addressing the following comments.

Major Remark

=====

1- Making a statement about causality based on observing correlation or similarity is not valid, while the reverse is valid. Therefore, on page 6, Lines 14-17, the statement "In other words, the major driving forces behind the organization of these networks are similar. Made simple by example, if the triadic closure is a driving force behind the organization of Advice network then it is very likely that the same holds for Discuss layer." is not valid. This statement seems intuitively valid, but scientifically is wrong. According to the context of the paper, it is recommended that limit the conclusions in the level of the similarity and do not generalize into the concept of causality.

Minor Remarks

=====

2- In Figure 3, for $K=1$ (Left figure), it can be observed overall that AUROC for LRM increases with the fraction of removed links. Please explain this observation (as we expect intuitively to see a reduction in AUROC by increasing the fraction of removed links)?

3- In Fig. 4, we see three figures corresponding to three different values of K and we still do not know the optimum value of K and we can see just the trend of the changes. It is recommended to have just one figure showing AUROC for a fixed fraction of removed links vs. different values of K . We can have multiple curves for different values of the fraction of removed links.

Grammar & Typos

=====

4-Please read the paper to correct some grammar mistakes.

5- Punctuations need to be revised (e.g., when you have "... A,B, and C..." you need comma before "and". For example, on page 2 lines 36-38 you have missed comma before "and": Real world networks like biochemical, human (HERE) and air transportation networks are examples of biological, social (HERE) and technological systems, respectively.). In many places, commas are missed. For example, on page 2 line 46, "In a similar way, (YOU NEED COMMA HERE) people are ...".

6- Page 6, Line 50: "The results of Table 1 indicates that" -> "The results of Table 1 indicate that"

7- Page 13, Line 30: "...can be clarified more using results in Figure 4." -> "can be clarified more by using the results depicted in Figure 4."

Author's Response to Decision Letter for (RSOS-191928.R0)

See Appendix A.

RSOS-191928.R1 (Revision)

Review form: Reviewer 1

Is the manuscript scientifically sound in its present form?

Yes

Are the interpretations and conclusions justified by the results?

Yes

Is the language acceptable?

Yes

Do you have any ethical concerns with this paper?

No

Have you any concerns about statistical analyses in this paper?

Yes

Recommendation?

Accept as is

Comments to the Author(s)

The manuscript has been properly revised to address the comments. I support the acceptance.

Review form: Reviewer 2

Is the manuscript scientifically sound in its present form?

Yes

Are the interpretations and conclusions justified by the results?

Yes

Is the language acceptable?

Yes

Do you have any ethical concerns with this paper?

No

Have you any concerns about statistical analyses in this paper?

No

Recommendation?

Accept with minor revision (please list in comments)

Comments to the Author(s)

Thanks for all the changes and comments. I just saw a typo in the caption of Fig. 4:

... The result indicates the ( it seems it should be "that" rather than "the) using (add "an") appropriate number of leading eigenvectors leads to

Decision letter (RSOS-191928.R1)

Dear Mr Abdolhosseini-Qomi:

On behalf of the Editors, I am pleased to inform you that your Manuscript RSOS-191928.R1 entitled "Link Prediction in Real-World Multiplex Networks via Layer Reconstruction Method" has been accepted for publication in Royal Society Open Science subject to minor revision in accordance with the referee suggestions. Please find the referees' comments at the end of this email.

The reviewers and Subject Editor have recommended publication, but also suggest some minor revisions to your manuscript. Therefore, I invite you to respond to the comments and revise your manuscript.

- Ethics statement

- Data accessibility

<http://datadryad.org/submit?journalID=RSOS&manu=RSOS-191928.R1>

- Competing interests

- Authors' contributions

- Acknowledgements

- Funding statement

Because the schedule for publication is very tight, it is a condition of publication that you submit the revised version of your manuscript before 19-Jun-2020. Please note that the revision deadline will expire at 00.00am on this date. If you do not think you will be able to meet this date please let me know immediately.

Supplementary files will be published alongside the paper on the journal website and posted on the online figshare repository (<https://figshare.com>). The heading and legend provided for each supplementary file during the submission process will be used to create the figshare page, so please ensure these are accurate and informative so that your files can be found in searches. Files

on figshare will be made available approximately one week before the accompanying article so that the supplementary material can be attributed a unique DOI.

on behalf of Professor Matjaz Perc (Associate Editor) and Miles Padgett (Subject Editor)
openscience@royalsociety.org

Reviewer comments to Author:
Reviewer: 1

Comments to the Author(s)
The manuscript has been properly revised to address the comments. I support the acceptance.

Reviewer: 2

Comments to the Author(s)
Thanks for all the changes and comments. I just saw a typo in the caption of Fig. 4:

... The result indicates the ( it seems it should be "that" rather than "the) using (add "an") appropriate number of leading eigenvectors leads to

Author's Response to Decision Letter for (RSOS-191928.R1)

See Appendix B.

Decision letter (RSOS-191928.R2)

Dear Mr Abdolhosseini-Qomi,

It is a pleasure to accept your manuscript entitled "Link Prediction in Real-World Multiplex Networks via Layer Reconstruction Method" in its current form for publication in Royal Society Open Science.

You can expect to receive a proof of your article in the near future. Please contact the editorial office (openscience_proofs@royalsociety.org) and the production office

(openscience@royalsociety.org) to let us know if you are likely to be away from e-mail contact -- if you are going to be away, please nominate a co-author (if available) to manage the proofing process, and ensure they are copied into your email to the journal.

Kind regards,
Lianne Parkhouse
Editorial Coordinator
Royal Society Open Science
openscience@royalsociety.org

on behalf of Professor Matjaz Perc (Associate Editor) and Miles Padgett (Subject Editor)
openscience@royalsociety.org

Appendix A

Response to the comments of the reviewers of
Royal Society Open Science
on the manuscript entitled

**“Link Prediction in Real-World Multiplex Networks via Layer Reconstruction
Method”**

(Submission ID: RSOS-191928)

Dear Editor:

We thank you and the reviewers for their helpful comments and suggestions, according to which we have revised the manuscript. In what follows, we state the changes made in the paper and reply to each comment. In the revised version, all changes in the text are shown in blue. In this response letter, comments of the reviewers are reflected in the **typewriter** form and our replies are reported in the ordinary print style.

After addressing the issues raised, we hope that the concerns of the reviewers have been obviated and you find our manuscript suitable for publication.

Amir Mahdi Abdolhosseini-Qomi (Corresponding Author)

Response to Reviewer 1

This manuscript has proposed a new link prediction method for multiplex networks. The results are interesting and I do support the acceptance provided that the authors address the following comments:

We would like to thank the esteemed reviewer for his/her constructive comments.

Comment:

There is mix of concepts in the abstract. While the authors refer to heterogeneity of entities, they immediately mention multiplex networks. In multiplex networks, all the nodes are from the same type, but link could be developed in different layers. I suggest removing the first.

✓**Response:** We understand your concern. The concept of multiplex networks is discussed in the abstract emphasizing on different types of links and not nodes as we can see in the beginning of the abstract:

“Networks are invaluable tools to study real biological, social and technological complex systems in which connected elements form a purposeful phenomenon. A higher resolution image of these systems shows that the connection types do not confine to one but to a variety of types. Multiplex networks encode this complexity with a set of nodes which are connected in different layers via different types of links.”

Comment:

Writing needs significant improves and the manuscript should go under a careful proofreading. The fonts on the math within text is much larger than the the rest of the text, which make it difficult to follow them. My advice is to convert everything to LaTeX.

✓**Response:** As you suggested, we have re-written the whole manuscript using L^AT_EX and a rigorous phase of proof-reading has been conducted.

Comment:

Eigenvectors of an adjacency matrix contain only some of its structural information, e.g. one can partition the network based on the eigenvector corresponding to the largest eigenvalue.

✓Response:

1. We approve that eigenvectors do not contain all of the structural information, at least dropping the eigenvalues neglects a part of information. As the adjacency matrix is diagonalizable, it can be stated that all of the eigenvalues and eigenvectors together, contain all of the information of the corresponding adjacency matrix.
2. But, eigenvectors corresponding to lower eigenvalues contain information at different scales.

More on 2:

So far, most of the layer similarity measures introduced in the literature [1, 2, 3] examine the structure of layers at a specific scale. This is despite the fact that the main characteristics of complex networks is properly described at all scales ranging from microscales (node properties) and mesoscales (communities at different resolutions) to

macroscales (whole network properties) [4]. It has been shown that the spectral information of the Laplacian matrix reveals the structural properties at different scales, (i) the number of null eigenvalues gives trivially the number of disconnected components, (ii) the gaps between consecutive eigenvalues tell us about the hierarchical structure, and (iii) large eigenvalues stand for the existence of hubs in the network [4, 5, 6].

In [7], authors have given a concrete example considering a real-world network, the Santa Fe Institute (SFI) collaboration network [8]. They have shown that how clusters of larger scales can be identified by smaller eigenvalues and their corresponding eigenvectors of the Laplacian matrix and vice versa.

It should be noted that in contrast to Laplacian matrix, the large eigenvalues and their corresponding eigenvectors of adjacency matrix relates to large scale properties of the network and vice versa.

Therefore, we use the eigenvectors of the adjacency matrix as representatives of structural features to compare the layers of the multiplex networks.

A summary of the above discussion is added at page 4, at the beginning of Section 3.1.

Comment:

I am still not clear why similarity between the eigenvectors should be meaningful; detailed discussion is required for that. They authors may discuss relevance of these similarities to the existing literature, e.g. doi.org/10.1098/rsos.171747.

✓Response:

The following paragraphs clarify the meaning of eigenvectors similarity. A summary of these paragraphs are presented at Section 3.1:

Here, we study the similarity of layers in multiplex networks with each layer being a simple graph. One way to represent a simple graph is via the adjacency matrix that contains all information of the corresponding layer. The structure of each layer is made of some substructures of local (e.g. triads) and global (e.g. hubs) importance. Here, we state that a good layer similarity measure should compare layers based on all substructures, from local to global ones. Eigenvectors of adjacency matrix is known to be associated with substructures of networks [9, 10, 7, 4]. We introduce a new measure for similarity of layers in multiplex networks which is different from the existing literature [3] as it takes various substructures into account at the same time.

Relevance to existing literature:

A systematic study of measures for quantifying layer similarity has been proposed in [3]. This work recommends network representation using property matrices. A property matrix \mathbf{P} is a matrix where:

- the columns correspond to a set \mathbf{S} of network structures (nodes, edges, triangles,...),
- the rows correspond to a set \mathbf{C} of contexts where these structures are observed (layers, groups, snapshots,...),
and
- $p_{s,c}$ is the value of an observational function mapping each pair structure/context into a number (degree, distance,...).

The property matrices introduced in [3] consider each layer as a context. Then the occurrences of different structures in each layer is reported by values of observational function or layer property vector. Using eigenvectors for comparison of layers means that we are comparing them at different scales. So, here each context refers to measuring each layer at a specific scale. Having nodes as network structure, each property vector contains the information of node activity at specific scale of study. As we have numerical property vectors (eigenvector entry for each node), using cosine similarity is one of the options as mentioned in [3]. Therefore, in this way we are able to compare the structure of layers in a multi-scale manner and report a mixture of similarities between layers.

Comment:

Figure captions should be extended by including further details. This will make them more readable.

✓**Response:** Now, all Figures and Tables have extended description.

Comment:

The results have been compared with only a single baseline method, which is a rather old publication (2015). The authors should compare the performance with more state-of-the methods, e.g. doi.org/10.1093/comnet/cnz007. I also noticed that some of the references cited in the manuscript have also proposed link prediction methods for multiplex networks. They also need to be compared with.

✓**Response:** We added a section for comparison with the state-of-the-art methods and an extensive comparison table, Section 3.5 and Table 2. We considered recent works of [1] and [11]. The total number of 29 duplex networks are considered for this study. We compared the total of three methods (LRM, Perturbed-LRM, Perturbed-LRM-GOR) of our own and 10 methods from the literature (top 4 performing are reported for succinctness). Our proposed Perturbed-LRM-GOR is the top performing as it wins in all pairwise comparison against rival methods.

Response to Reviewer 2

The work has proposed an approach for link prediction in Multiplex Networks and it has illustrated the benefit of applying multiple layers in link prediction compared to using just a single layer. The paper may be accepted subject to appropriate revision addressing the following comments.

We would like to thank the esteemed reviewer for his/her constructive comments.

Comment:

Major Remark

1- Making a statement about causality based on observing correlation or similarity is not valid, while the reverse is valid. Therefore, on page 6, Lines 14-17, the statement "In other words, the major driving forces behind the organization of these networks are similar. Made simple by example, if the triadic closure is a driving force behind the organization of Advice network then it is very likely that the same holds for Discuss layer." is not valid. This statement seems intuitively valid, but scientifically is wrong. According to the context of the paper, it is recommended that limit the conclusions in the level of the similarity and do not generalize into the concept of causality.

✓**Response:** We confirm that the our experiments do not contain the required evidence for inference of causality of link formation between layers. However, the basis of our method is correlated presence of the links between layers of multiplex networks which leads to enhancement in link prediction performance. We corrected our wording to avoid confusion in this regard. The revised phrasing could be seen at Page 5, Section 3.2.

Comment:

Minor Remarks

2- In Figure 3, for $K=1$ (Left figure), it can be observed overall that AUROC for LRM increases with the fraction of removed links. Please explain this observation (as we expect intuitively to see a reduction in AUROC by increasing the fraction of removed links)?

✓**Response:** A slight increase of performance at the high fraction of removed links is a special case for this synthetic network. Large fraction of removed links leads to smaller magnitude of scores provided by LRM(Self). As the information transferred from auxiliary layer is rich (LRM(backup) as in Figure R1), the addition of LRM(Self) degrades the performance. This degradation of performance is less for smaller magnitude of LRM(Self) scores, so it seems that the overall performance is increasing. A summary of this paragraph is presented in Page 12, Paragraph 5.

Comment:

Minor Remarks

3- In Fig. 4, we see three figures corresponding to three different values of K and we still do not know the optimum value of K and we can see just the trend of the changes. It is recommended to have just one figure showing AUROC for a fixed fraction of removed links vs. different values of K . We

Fig. R1: The result of Figure 3 of paper when the contribution of target layer is removed from LRM.

can have multiple curves for different values of the fraction of removed links.

✓**Response:** We replaced that figure according to your suggestion. The new Figure 4 at Page 13 is now according to this suggestion.

Comment:

Grammar & Typos

4-Please read the paper to correct some grammar mistakes.

✓**Response:** A rigorous phase of proof-reading has been conducted and these mistakes, along with others, have been corrected.

Comment:

Grammar & Typos

5- Punctuations need to be revised (e.g., when you have "... A,B, and C..." you need comma before "and". For example, on page 2 lines 36-38 you have missed comma before "and": Real world networks like biochemical, human (HERE) and air transportation networks are examples of biological, social (HERE) and technological systems, respectively.). In many places, commas are missed. For example, on page 2 line 46, "In a similar way, (YOU NEED COMMA HERE) people are ...".

✓**Response:** A rigorous phase of proof-reading has been conducted and these mistakes, along with others, have been corrected.

Comment:

Grammar & Typos

6- Page 6, Line 50: "The results of Table 1 indicates that" -> "The results of Table 1 indicate that"

✓**Response:** A rigorous phase of proof-reading has been conducted and these mistakes, along with others, have been corrected.

Comment:

Grammar & Typos

7- Page 13, Line 30: "...can be clarified more using results in Figure 4." -> "can be clarified more by using the results depicted in Figure 4."

✓**Response:** A rigorous phase of proof-reading has been conducted and these mistakes, along with others, have been corrected.

References

- [1] Y. Yao, R. Zhang, F. Yang, Y. Yuan, Q. Sun, Y. Qiu, and R. Hu, “Link prediction via layer relevance of multiplex networks,” *International Journal of Modern Physics C*, vol. 28, no. 08, p. 1750101, 2017.
- [2] M. De Domenico, V. Nicosia, A. Arenas, and V. Latora, “Structural reducibility of multilayer networks,” *Nature communications*, vol. 6, no. 1, pp. 1–9, 2015.
- [3] P. Bródka, A. Chmiel, M. Magnani, and G. Ragozini, “Quantifying layer similarity in multiplex networks: a systematic study,” *Royal Society open science*, vol. 5, no. 8, p. 171747, 2018.
- [4] A. Arenas, A. Díaz-Guilera, and C. J. Pérez-Vicente, “Synchronization reveals topological scales in complex networks,” *Physical review letters*, vol. 96, no. 11, p. 114102, 2006.
- [5] L. Donetti and M. A. Munoz, “Detecting network communities: a new systematic and efficient algorithm,” *Journal of Statistical Mechanics: Theory and Experiment*, vol. 2004, no. 10, p. P10012, 2004.
- [6] A. Capocci, V. D. Servedio, G. Caldarelli, and F. Colaiori, “Detecting communities in large networks,” *Physica A: Statistical Mechanics and its Applications*, vol. 352, no. 2-4, pp. 669–676, 2005.
- [7] G.-M. Zhu, H. Yang, R. Yang, J. Ren, B. Li, and Y.-C. Lai, “Uncovering evolutionary ages of nodes in complex networks,” *The European Physical Journal B*, vol. 85, no. 3, p. 106, 2012.
- [8] M. Girvan and M. E. Newman, “Community structure in social and biological networks,” *Proceedings of the national academy of sciences*, vol. 99, no. 12, pp. 7821–7826, 2002.
- [9] A. E. Brouwer and W. H. Haemers, *Spectra of graphs*. Springer Science & Business Media, 2011.
- [10] G. Zhu, H. Yang, C. Yin, and B. Li, “Localizations on complex networks,” *Physical Review E*, vol. 77, no. 6, p. 066113, 2008.
- [11] Z. Samei and M. Jalili, “Application of hyperbolic geometry in link prediction of multiplex networks,” *Scientific reports*, vol. 9, no. 1, pp. 1–11, 2019.

Appendix B

The second round of response to *Royal Society Open Science* reviewers' comments on the manuscript entitled

“Link Prediction in Real-World Multiplex Networks via Layer Reconstruction Method”

(Submission ID: RSOS-191928)

Dear Editor:

We thank you and the reviewers for their helpful comments and suggestions, according to which we have revised the manuscript.

Amir Mahdi Abdolhosseini-Qomi (Corresponding Author)

Response to Reviewer 1

The manuscript has been properly revised to address the comments. I support the acceptance.

We appreciate his/her comments and supports.

Response to Reviewer 2

Thanks for all the changes and comments. I just saw a typo in the caption of Fig. 4:

... The result indicates the (it seems it should be "that" rather than "the") using (add "an") appropriate number of leading eigenvectors leads to

We appreciate his/her comments and suggestions. The revised sentence is as follows:

The result indicates that using an appropriate number of leading eigenvectors leads to good performance while minimizes the susceptibility to noise-like patterns.